# EMBO *reports*

# Dual-color live imaging unveils stepwise organization of multiple basal body arrays by cytoskeletons

Gen Shiratsuchi[1,2,14], Satoshi Konishi [ID][2,3,4,5,14], Tomoki Yano[2,6], Yuichi Yanagihashi[7],
Shogo Nakayama [ID][2,8], Tatsuya Katsuno [ID][2,9], Hiroka Kashihara[1,2], Hiroo Tanaka[1,2,10], Kazuto Tsukita [ID][1,2,11],
Koya Suzuki [ID][1,2], Elisa Herawati [ID][2,12], Hitomi Watanabe[13], Toyohiro Hirai[3], Takeshi Yagi[2], Gen Kondoh[13],
Shimpei Gotoh [ID][3,4], Atsushi Tamura[1,2,10]✉ & Sachiko Tsukita [ID][1,2]✉

## Abstract

For mucociliary clearance of pathogens, tracheal multiciliated epithelial cells (MCCs) organize coordinated beating of cilia, which originate from basal bodies (BBs) with basal feet (BFs) on one side. To clarify the self-organizing mechanism of coordinated intracellular BB-arrays composed of a well-ordered BB-alignment and unidirectional BB-orientation, determined by the direction of BB to BF, we generated double transgenic mice with GFP-centrin2-labeled BBs and mRuby3-Cep128-labeled BFs for long-term, high-resolution, dual-color live-cell imaging in primary-cultured tracheal MCCs. At early timepoints of MCC differentiation, BB-orientation and BB-local alignment antecedently coordinated in an apical microtubule-dependent manner. Later during MCC differentiation, fluctuations in BB-orientation were restricted, and locally aligned BB-arrays were further coordinated to align across the entire cell (BB-global alignment), mainly in an apical intermediate-sized filament-lattice-dependent manner. Thus, the high coordination of the BB-array was established for efficient mucociliary clearance as the primary defense against pathogen infection, identifying apical cytoskeletons as potential therapeutic targets.

**Keywords** Apical Cytoskeletons; Ciliary Basal Bodies; Coordinated Ciliary Beating; Mucociliary Clearance; Multiciliated Cells
**Subject Category** Cell Adhesion, Polarity & Cytoskeleton

## Introduction

Mammalian tracheal multiciliated epithelial cells (MCCs) contain multiple cilia with unidirectionally coordinated beating to generate mucociliary clearance flow, and this plays essential roles in host defense, particularly in the elimination of microorganisms including viruses, and foreign substances (Wanner et al, 1996; Knowles and Boucher, 2002; Fahy and Dickey, 2010; Tilley et al, 2015). Aberrant mucociliary clearance flow is also associated with airway diseases, such as cystic fibrosis, primary ciliary dyskinesia, bronchiectasis, asthma, and chronic obstructive pulmonary disease (Tilley et al, 2015; Ratjen et al, 2015; Shapiro et al, 2018; Boucher, 2019; Chivukula et al, 2020; Legendre et al, 2021). Therefore, for pathological characterization and identification of therapeutic targets in these diseases, it is crucial to better understand the mechanisms behind the coordinated beating of multiple cilia. Hence, elucidation of the underlying mechanisms of mucociliary clearance flow formation is an urgent issue.

In MCCs, hundreds of cilia are finely associated with the apical membranes for intracellular and intercellular coordination of ciliary beating (Gibbons, 1961; Reed et al, 1984; Wanner et al, 1996; Brooks and Wallingford, 2014). The axonemes of the ciliary shafts extend from basal bodies (BBs), which anchor to the apical membrane. BBs have appendage-like structures called basal feet (BFs), associated with BBs on one side at positions 4, 5, and 6 of circularly arranged triplets of microtubules (MTs) (Sorokin, 1968; Anderson, 1972; Kunimoto et al, 2012; Clare et al, 2014). The BB-orientation, determined by the direction of BBs to BFs, defines the direction of ciliary beating. Thus, loss of the basal foot (BF) disrupts the coordination of the direction of ciliary beating (Kunimoto et al, 2012). Hence, the critical roles of BFs in the determination of BB-orientation are a major focus of many researchers.

The BB-orientation is well coordinated both intra- and intercellularly via planar cell polarity (PCP) during the differentiation of MCCs (Park et al, 2008; Guirao et al, 2010; Tissir et al, 2010; Wallingford, 2010; Vladar et al, 2012; Boutin et al, 2014; Takagishi et al, 2017; Robinson et al, 2020; Nakayama et al, 2021; Usami et al, 2021). Among PCP-associated cues, apical cytoskeletons, apical actin filaments, and MTs, are associated with ciliary BBs in the

[1]Advanced Comprehensive Research Organization, Teikyo University, Tokyo, Japan. [2]Graduate School of Frontier Biosciences, Osaka University, Osaka, Japan. [3]Department of Respiratory Medicine, Graduate School of Medicine, Kyoto University, Kyoto, Japan. [4]Center for iPS Cell Research and Application, Kyoto University, Kyoto, Japan. [5]Department of Cell Biology, Duke University School of Medicine, Durham, NC, USA. [6]Department of Organoid Medicine, Sakaguchi Laboratory, Keio University School of Medicine, Tokyo, Japan. [7]THINKCYTE, Inc., Tokyo, Japan. [8]RIKEN Center for Biosystems Dynamics Research, Hyogo, Japan. [9]Center for Anatomical Studies, Graduate School of Medicine, Kyoto University, Kyoto, Japan. [10]School of Medicine, Teikyo University, Tokyo, Japan. [11]Department of Neurology, Graduate School of Medicine, Kyoto University, Kyoto, Japan. [12]Faculty of Mathematics and Natural Sciences, Universitas Sebelas Maret, Surakarta, Central Java, Indonesia. [13]Institute for Life and Medical Sciences, Kyoto University, Kyoto, Japan. [14]These authors contributed equally: Gen Shiratsuchi, Satoshi Konishi. ✉E-mail: atamura@med.teikyo-u.ac.jp; atsukita@med.teikyo-u.ac.jp; stsukitatjcl@gmail.com

apical plane in MCCs (Wallingford, 2010; Werner et al, 2011; Kunimoto et al, 2012; Antoniades et al, 2014; Herawati et al, 2016; Tateishi et al, 2017). Actin filaments regulate proper ciliogenesis and organization of ciliary BBs in MCCs of mouse oviducts, brain ventricles, and the *Xenopus* epidermis (Hirota et al, 2010; Werner et al, 2011; Fuertes-Alvarez et al, 2018; Mahuzier et al, 2018). BBs and BFs are spatiotemporally associated with apical MTs, which show a biased distribution along PCP and accumulate in the Frizzed-Daple side to coordinate BB-orientation in MCCs of mouse tracheas, ependymal cells, and *Xenopus* epidermal cells (Vladar et al, 2012; Herawati et al, 2016; Takagishi et al, 2017; Kim et al, 2018; Nakayama et al, 2021). In addition to apical actin filaments and MTs, apical localization of intermediate-sized filaments (IFs) surrounding ciliary BBs has been reported in mouse tracheal MCCs (Herawati et al, 2016; Tateishi et al, 2017); however, its function and relationship to coordination of the BB-array in MCCs still remain unclear. The fluid flow created by multiple motile cilia has been proposed as an essential factor in the polarization of MCCs with PCP signals, based on observations in the Xenopus epidermis (Mitchell et al, 2007; Mitchell et al, 2009; Werner and Mitchell, 2012) and mouse ependyma (Guirao et al, 2010), and human airways (Biggart et al, 2001). These reports strongly suggest a positive feedback mechanism of the BB-array organization.

In our previously reported long-term high-resolution live-cell imaging system of BBs, we used an air-liquid interface (ALI) primary culture system for mouse tracheal epithelial cells (MTECs) to recapitulate the differentiation of MCCs from airway basal stem cells that undergo de novo motile ciliogenesis (Davidson et al, 2000; You et al, 2002). In this system, centrin2 was labeled with green fluorescent protein (GFP). Based on live-cell images of BBs, we previously defined the linearly ordered alignment of BBs as BB-alignment and included a description of BB-orientation; using these parameters, we showed that the differentiation of the BB-alignment pattern was highly dynamic and was achieved through four stereotypical patterns, i.e., "floret," "scatter," "partial alignment," and "alignment" (Herawati et al, 2016). Subsequently, we considered that the regular BB-array observed at the apical side of the mouse tracheal MCCs involves both BB-orientation and BB-alignment; however, the real-time relationships between these features and the critical roles of BB-alignment in coordinated ciliary beating remain unclear.

Accordingly, in this study, we developed a novel dual-color fluorescence system for long-term, high-resolution live-cell imaging of GFP-tagged BBs and mRuby3-tagged BFs in MTECs. This system is a significantly improved version compared to our previous methods and can reveal the real-time relationships between BB orientation and alignment. In addition, we introduced the concept of "global BB-alignment" in mature MCCs which differs from the "local BB-alignment," a roughly synonymous term with "BB-alignment" used in our previous study. Previously we focused on the formation of locally aligned shorter lines of BBs during the early stages of MCC differentiation (3–12 days of ALI culture for MTECs) (Herawati et al, 2016). By contrast, in our attempt to assess the physiological relevance of BB-alignment, we incorporated the final differential stages (after 13 days of ALI culture) of "BB-global alignment," considering the directional coordination of locally aligned shorter BB lines, as is observed in the mouse trachea in vivo. Using a new dual-color fluorescence live-cell imaging and classification method, in this study, we elucidated the stepwise organization of apical MTs and IFs in BB-arrays, demonstrating the initial lead role of apical MTs for the formation of BB-orientation and BB-local alignment and the following role of apical IFs for the establishment of BB-global alignment. We also identified the roles of the apical cytoskeleton in the coordination of BB-global alignment to facilitate efficient mucociliary clearance for prevention of pathogen infection.

## Results

### Development of a dual-color fluorescence live-cell imaging system for BBs and BFs in MCCs of MTECs prepared from GFP-centrin2 and mRuby3-Cep128 double-transgenic mice

In our new system for long-term, high-resolution live-cell imaging with dual-color fluorescence, we analyzed BB-orientation and BB-alignment simultaneously for differently colored BBs and BFs in MCCs of MTECs (Appendix Fig. S1A,B). This represents obvious progression from single-color live-cell imaging systems in which only BBs, not BFs, have been live imaged after centrosome targeting to the apical membranes, revealing their highly regular alignment (Herawati et al, 2016). We previously found four stereotypical patterns of BB-local alignment, without assessment of the dynamic relationship with BB-orientation and the directional coordination of locally formed shorter lines of BBs (Appendix Fig, S1C) (Herawati et al, 2016). To answer these remaining questions, BB-orientation and BB-alignment should be simultaneously detected in real time. Thus, we generated transgenic mice in which BBs and BFs were double-labeled. We attempted to generate transgenic mice using fluorescence-tagged BF components, such as centriolin (Gromley et al, 2003) and Cep128 (Mazo et al, 2016; Monnich et al, 2018; Kashihara et al, 2019; Chong et al, 2020), in GFP-centrin2 transgenic mice (Higginbotham et al, 2004). Owing to the brightness and photostability of the fluorescence under dual-color laser irradiation, we eventually decided to use mRuby3-Cep128 (Bajar et al, 2016) for generation of transgenic mice. Notably, Cep128 stands out as a protein that localizes to the centriolar subdistal appendages (Mazo et al, 2016; Monnich et al, 2018; Kashihara et al, 2019) and prominently marks the BF of ciliated BBs (Nguyen et al, 2020). Loss of Cep128 impaired localization of other BF components such as centriolin, ninein, and Cep170 (Mazo et al, 2016; Kashihara et al, 2019). Given these compelling findings, we concluded that Cep128 should be the quintessential marker for functional BF in our study. We then crossed these mice with GFP-centrin2 transgenic mice to generate double-labeled transgenic mice harboring GFP-centrin2-labeled BBs and mRuby3-Cep128-labeled BFs (Fig. 1A–C). When we compared the locations of the fluorescent signals of centrin2 and Cep128 with the immunofluorescent signal of centriolin, another BF marker, in MCCs of MTECs derived from the double-labeled transgenic mice, we confirmed the appropriate locations of fluorescent signals of centrin2, Cep128, and centriolin in this order (Fig. 1C,D; Appendix Fig. S1D) (Mazo et al, 2016; Kashihara et al, 2019). Here, we successfully generated double-transgenic mice in which the BBs and BFs were doubly live-labeled with GFP-centrin2 and mRuby3-Cep128, respectively.

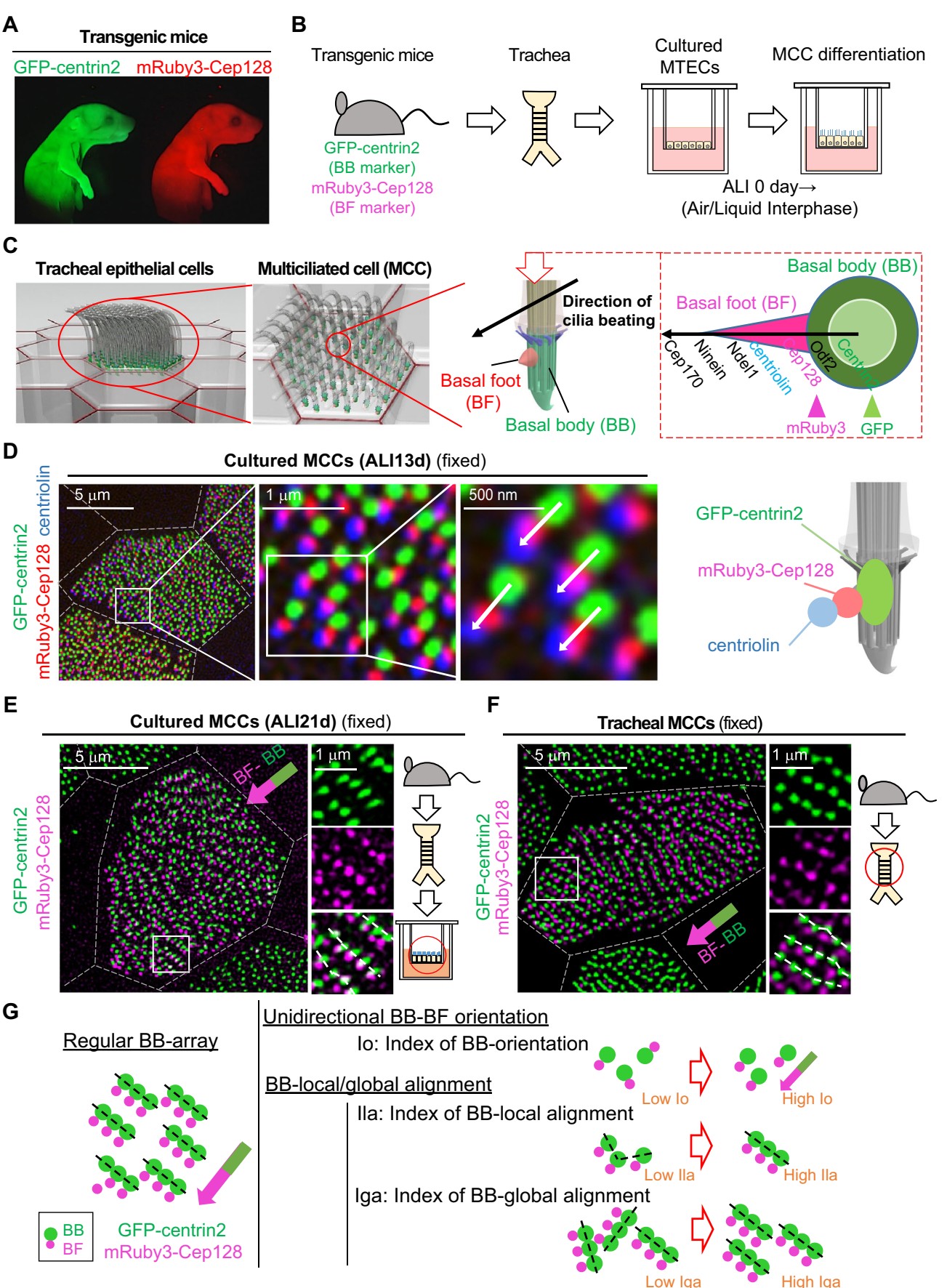

**Figure 1.  Establishment of a high-resolution, dual-color live-cell imaging system for analysis of the BB-array.**

(A) Transgenic mice expressing GFP-centrin2 (green) and mRuby3-Cep128 (red). (B) Schematic diagrams of our procedure for the preparation of MTECs from transgenic mice expressing GFP-centrin2 and mRuby3-Cep128. (C) Schematic diagrams of cilia/BBs/BFs in MCCs of MTECs. Mature tracheal MCCs with cilia/BBs/BFs aligned in a linear line and a magnified illustration of a cilium/BB/BF complex are shown. A horizontal view of a BB/BF complex with major known components is also shown (right panel). We used GFP-centrin2 and mRuby3-Cep128 as markers of BBs and BFs in our assay. (D) Fixed MTECs (ALI13d), which were prepared from transgenic mice expressing GFP-centrin2 (green) and mRuby3-Cep128 (red), were stained with antibodies against centriolin (blue) and analyzed by spinning disk confocal microscopy. Unidirectional BB-orientation (white arrows) and beautifully ordered alignments of three foci in MCCs are shown. Bars, 5 μm, 1 μm, and 500 nm. Gray dotted lines represent the cell shapes estimated from the background signals. Schematic diagrams of BB/BF and the markers are also shown (right panel). (E,F) Spinning disk confocal microscopy of the BB-array in MCCs of fixed mature MTECs (ALI21d) (E) and tracheal epithelial cells (F) prepared from transgenic mice expressing GFP-centrin2 (green) and mRuby3-Cep128 (magenta). Unidirectional BB-orientation (arrows) and ordered BB-alignment were clearly observed in both MCCs of mature MTECs (E) and tracheal cells (F). Insets show 2.5- and 2.0-fold magnified images of fluorescent foci. Bars, 5 μm and 1 μm. Gray dotted lines represent the cell shapes estimated from the background signals. White dotted lines in insets represent the aligned shorter lines of BBs. Schematic diagrams of the sample preparation are also shown (right panels). (G) Schematic diagrams of the indexes used in this study. Io (index of orientation) was used to quantify the degree of unidirectional BB-orientation. Ila (index of local alignment) and Iga (index of global alignment) were used to quantify the degrees of BB-alignments, within a small BB group (Ila) and in a whole cell (Iga). Source data are available online for this figure.

Moreover, a question raised previously was that in BB-local alignment, the directional coordination of locally formed shorter lines of BBs was not noted before the final differentiation of MCCs, at 3–12 days after conventional medium culture was switched to ALI culture (Fig. 1G; Appendix Fig. S1C) (Herawati et al, 2016). In the current study, we examined the nearly final differentiation of the regular BB-array in MCCs of MTECs because the well-developed linear BB-global alignment appeared after 13 days of ALI culture (Fig. 1E; Appendix Fig. S1E), resembling that of in vivo tracheal MCCs prepared from adult mice (Fig. 1F; Appendix Fig. S1F). We defined the index of BB-global alignment (Iga) to judge how the direction of each aligned shorter BB line was coordinated globally, taking the directional coordination of locally formed shorter lines of BBs into consideration (Fig. 1G; Appendix Figs. S1C, S2).

Overall, we performed dual-color live-cell imaging of BBs of MCCs with a focus on BB-orientation, BB-local alignment, and BB-global alignment. We followed the pattern of extreme maturation of BB-alignment by extending the culture time after 13 days ALI culture (Fig. 2).

## BB-orientation and BB-local alignment antecedently coordinated at early timepoints, and BB-global alignment subsequently coordinated during MCC maturation

Using dual-color live-cell imaging, we observed MTECs at 5–14 days ALI culture. The mRuby3-Cep128 signals were more unstable than GFP-centrin2 signals (Fig. EV1A, Movie EV1). Thus, we traced the dual fluorescent signals from both BBs and BFs for up to 48 h, with minimum beam damage. Subsequently, we followed the process of differentiation by connecting separate series of live-cell images on cells without beam damage, which enabled us to simultaneously analyze the developmental processes of BB-orientation, BB-local and BB-global alignment during tracheal MCC differentiation.

The coordination of BB-orientation was quantified as the index of orientation (Io), as previously described; high values were obtained when the BB-orientation was largely uniform in individual MCCs (Fig. 1G) (Herawati et al, 2016; Nakayama et al, 2021). To examine the relationships between the coordination of BB-orientation, BB-local alignment, and BB-global alignment, we performed continuous live-cell imaging (usually using 45-min intervals) for 6–48 h during the initial (5–6 days of ALI culture), early (7–12 days of ALI culture), and late (after 13 days of ALI

culture) timepoints (TPs) of MCC differentiation (Figs. 2A–E and EV1B). Our focus was on the ciliated BBs localized at the apical surface of tracheal MCCs, specifically after the time points during which deuterosome-mediated extensive centriole amplification was finalized and the cells became multiciliated (Zhao et al, 2013).

Live-cell imaging revealed that when BBs first appeared as floret patterns at the initial TPs, Io and Ila showed very low values (stage 1: floret stage, Io < 0.7, Ila < 0.23), as expected (Herawati et al, 2016). Subsequently, from initial to early TPs, we found that BB florets scattered over the apical cell plane (stage 2: scatter stage, Io < 0.7, 0.23 ≤ Ila < 0.33) and neither clustered nor formed regular alignments (Fig. 2B, Movie EV2). Next, from early to late TPs, 2–3 neighboring BBs appeared to align with each other in a side-by-side manner to form locally formed shorter lines of BBs (stage 3: partial alignment stage, Io < 0.7, 0.33 ≤ Ila), and these locally formed shorter lines of BB appeared to partly coalesce to form longer, parallel lines (stage 4: alignment pattern, 0.7 ≤ Io, Iga < 0.4; white dotted lines in Fig. 2C and Movie EV3). During these developmental stages at the early TP, Io and Ila were proportionally increased (Fig. EV1B).

From early to late TPs, although the BB-orientation was not significantly changed, Iga, which represents the degree of BB-global alignment, increased markedly, representing the final differentiation process of the BB-array (stage 5: global alignment pattern, 0.7 ≤ Io; 0.4 ≤ Iga; Fig. 2D,E, Movies EV4 and 5). Collectively, from initial to early TPs, Io and Ila increased in parallel, suggesting that BB-orientation and BB-local alignment may be spatiotemporally coordinated in MCCs at these TPs (Figs. 2F and EV1B). At late TPs, Io and Ila were unchanged, whereas Iga increased, suggesting that the BB-alignment was globally coordinated to progress to the final stage (Figs. 2F,G and EV1B). The biphasic patterns of the coordination in BB-orientation and BB-alignment, both local and global, were revealed by dual-color live-cell imaging of BBs and BFs.

Thus, we redefined the developmental stage of MCCs based on the index of local and global alignment (Fig. 2F,G). When Io was below 0.7, i.e., the boundary of the clusters of plots in Fig. 2F, we defined the stages by Ila, referring to our previous study, from the floret to partial alignment patterns (stages 1–3) (Herawati et al, 2016). In MCCs with an Io >0.7 and a high Iga (≥0.4) or low Iga (<0.4), the developmental stage was defined as stage 5, or stage 4, respectively. The biphasic developmental patterns of the BB-array, which included coordinated BB-orientation and BB-alignment, prompted us to propose a stepwise organization mechanism for the BB-array during tracheal MCC differentiation.

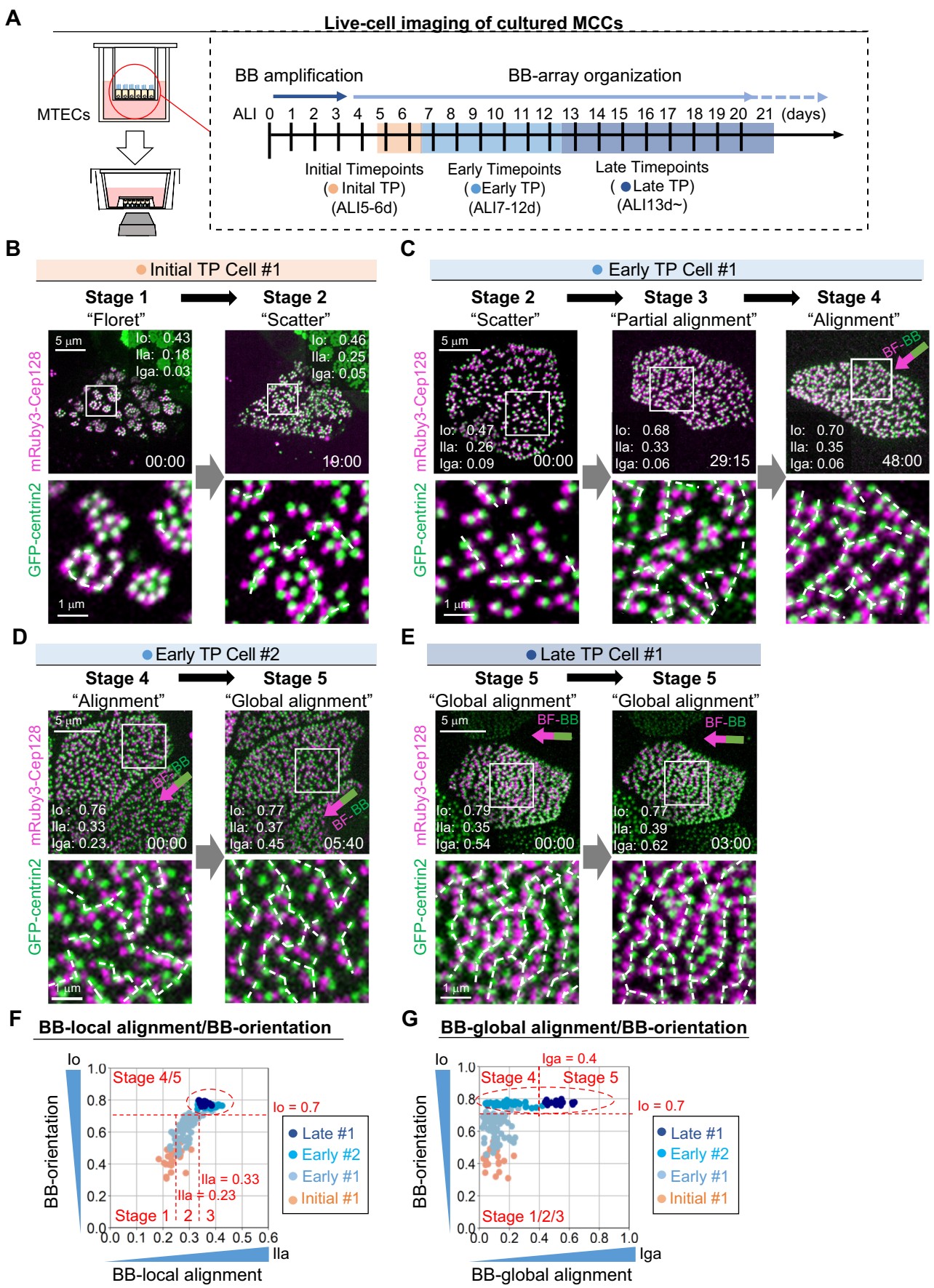

**Figure 2. Analysis and quantification of the developmental process of BB-arrays in mouse tracheal MCCs using high-resolution, dual-color live-cell imaging of MTECs.**

(A) Schematic diagram of the procedure for live analysis of the BB-array. MTECs prepared from transgenic mice expressing GFP-centrin2 and mRuby3-Cep128 were analyzed using our high-resolution, dual-color live-cell imaging system at "initial" (ALI5-6d), "early" (ALI7-12d), and "late" (ALI13d-) timepoints (TPs). (B-E) Dual-color live-cell imaging of MTECs prepared from transgenic mice expressing GFP-centrin2 (green) and mRuby3-Cep128 (magenta). Live-cell images of MCCs of MTECs at the initial TP (B, Initial TP Cell #1), early TP (C and D, Early TP Cell #1 and #2), and late TP (E, Late TP Cell #1) are shown with the values of Io, Ila, and Iga. We observed the transition from "floret" to "scatter" stage (B), from "scatter" to "alignment" stage (C), and from "alignment" to "global alignment" stage (D) as well as oscillations in the "global alignment" stage (E), using our dual-color live-cell imaging system. White dotted lines represent the alignments of adjacent BBs except for the ones aligned in a V-shape. Unidirectional BB-orientation (arrows) and ordered BB-alignment were observed (C-E). Insets show 5-/3.7-/3.3-/3.3-fold magnified images of fluorescent foci. Time is denoted in hours: minutes. Bars, 5 and 1 μm. See also Movies EV2-5. (F,G) Scatter plots of Io/Ila (F) and Io/Iga (G) in the MCCs shown in (B-E). Indexes acquired from all TPs of live-cell imaging are shown (Initial TP Cell #1: pink, Early TP Cell #1: sky blue, Early TP Cell#2: deep sky blue, Late TP Cell #1: navy blue). Newly defined developmental stages of the BB-array are also shown (stage 1: floret stage, stage 2: scatter stage, stage 3: partial alignment stage, stage 4: alignment pattern, stage 5: global alignment pattern). Line graphs of Io, Ila, and Iga are shown in Fig EV1B. Red dotted circles represent the plots from Early TP Cell #2 and Late TP Cell #1, which are clearly separated in scatter plots of Io/Iga (G). Red dotted lines represent Io = 0.7, Ila = 0.23/0.33, and Iga = 0.4. Source data are available online for this figure.

## Fixed samples of GFP-centrin2-labeled BBs and mRuby3-Cep128-labeled BFs confirmed the biphasic patterns of coordinated BB-orientation and BB-alignment

We next confirmed the biphasic development of BB-orientation and BB-alignment in many fixed MTEC samples without the concerns of possible beam damage in our live-cell imaging data (Figs. 3 and EV2). We examined the Io, Ila, and Iga in fixed MCCs in different stages (initial, early, and late TPs: $n = 16, 41, 56$) and performed statistical analyses. As a result, Io $(0.30 \pm 0.17, 0.49 \pm 0.25,$ and $0.84 \pm 0.07,$ respectively), Ila $(0.18 \pm 0.05, 0.34 \pm 0.07,$ and $0.45 \pm 0.05,$ respectively), and Iga $(0.15 \pm 0.07, 0.17 \pm 0.10,$ and $0.50 \pm 0.21,$ respectively) values were obtained for these TPs (Fig. EV2A,B). Thus, we confirmed the coordinated processes of BB-orientation, BB-local alignment, and BB-global alignment at each TP (Figs. 3B–E and EV2C,D). Statistical analyses showed no significant differences between Iga values in fixed MCCs at initial and early TPs, whereas Iga at the late TP was significantly increased from that at the early TP (Fig. EV2B).

By contrast, Ila and Io values showed continuous increases (Fig. EV2B), strongly supporting the biphasic patterns of the developmental processes of the BB-array predicted by our live-cell analysis. As shown in Fig. 3F,G, we concluded that the identification of stages (i.e., stages 1–5) was relevant because the Io, Ila, and Iga of fixed MCCs were consistent with those of live-cell imaged MCCs at each stage. Furthermore, we showed that after parallel development of BB-orientation and BB-local alignment, as shown by Io and Ila, in stages 1–4, the BB-global alignment was prominent in stage 5, i.e., the final stage of the differentiation of coordinated BB-arrays in MCCs.

## Fluctuation of BB-BF angles was restricted as MCC differentiation proceeded

Next, to examine BB-BF behaviors at each TP in detail, we conducted short-interval live-cell imaging at 10-min intervals for GFP-centrin2-labeled BBs and mRuby3-Cep128-labeled BFs and calculated the values of the indexes (Fig. 4). We assumed that MTECs at early TPs mainly included stages 1–4 MCCs, whereas those at late TPs mainly included stages 4–5 MCCs (Fig. 4A). Short-interval live-cell imaging analyses revealed that the locations and BB-BF angles were highly variable in stages 2 and 3 MCCs at early TPs (Fig. 4B, Movies EV6 and 7). In stage 4 MCCs at late TPs, the movement of BB-BF angles was clearly restricted. At early TPs, stage 4 MCCs showed relatively active movement of BB-BF angles,

as well as dynamic movement of locally formed shorter lines of BBs, as compared with those at late TPs (Fig. 4C, Movies EV8 and 9), suggesting the difference between the dynamics of BBs in MCCs at early and late TPs. Thus, when we used dual-color live-cell imaging to quantify the deviation of BB-BF angles, reflected by fluctuations in the BB-orientation, we found that dynamic BB-BF movement was significantly restricted in MCCs of mature MTECs around stages 4 and 5 at late TPs (Fig. 4D, BB-BF angle SD; $1.38 \pm 0.32$ at the early TP, $0.72 \pm 0.15$ at the late TP). Hence, stabilized movement of locally formed shorter lines of BBs and coordination of the BB-global alignment may have been closely related to restriction of the fluctuation of BB-BF angles. Accordingly, we then evaluated the mechanism mediating the decreased fluctuation of BB-BF angles and increased BB-global alignment in the sequential organization of the BB-array.

## Stepwise organization of apical MT and IF networks during MCC differentiation

Previously, apical cytoskeletons were identified just beneath the apical membranes of tracheal MCCs and shown to form layered meshwork structures surrounding BBs/BFs (Fig. 5A) (Tateishi et al, 2017; Yano et al, 2017). Hence, we examined the relationships of BB-arrays and the development of the apical cytoskeleton networks of MCCs of MTECs at initial, early, and late TPs. First, to investigate the localization patterns of apical MT and IF networks in tracheal MCCs, we performed immunostaining for α-tubulin and keratin8 as markers of apical MTs and IFs, which are highly expressed in mammalian airway multiciliated cells in tissue and culture (Fig. 5A–D) (Montoro et al, 2018; Ruiz Garcia et al, 2019). We observed the localization of these apical cytoskeletons at different TPs and analyzed differences in the signal densities of the cytoskeleton in projections of Z-stack images of the apical cell surface (Fig. 5B,C,E). At the initial TP, both MT and keratin8 networks were poorly formed. The density of apical MTs increased dramatically after the initial TP and relatively constant during the early/late TPs (Fig. 5B,E). We found that the density of the keratin8 meshwork was low in MCCs at the early TP and elevated at the late TP (Fig. 5C,E). Lattice-like structures of keratin8 networks were formed but sparse and uneven in the MCCs at the early TP. However, we frequently found well-organized lattice-like structures of keratin8 in mature MCCs of MTECs at the late TP (Fig. 5C). These lattice-like structures of the apical cytoskeleton networks seemed to surround individual BBs, as shown by the GFP-centrin2 signals (Fig. 5B–D).

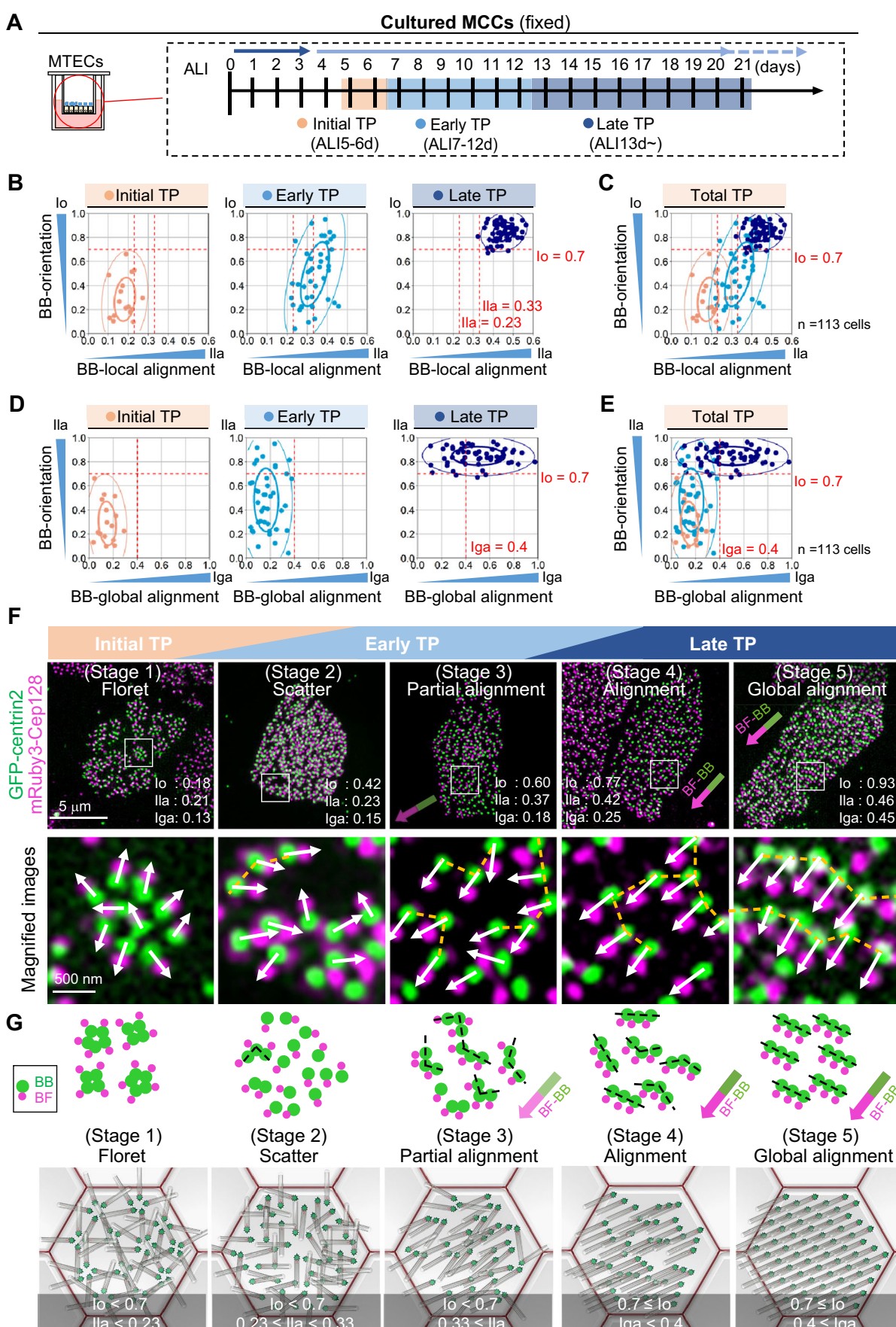

◀ **Figure 3.** Quantitative analysis of the developmental stages of the BB-array using fixed MTECs prepared at different TPs.

(**A**) Schematic diagrams of the procedure for the quantitative analysis of the BB-array using fixed MTECs prepared from transgenic mice expressing GFP-centrin2 and mRuby3-Cep128. (**B–E**) Scatter plots of Io/IIa (**B**) and Io/Iga (**D**) in MCCs of MTECs prepared from transgenic mice expressing GFP-centrin2 and mRuby3-Cep128 at the initial TP (pink, $n = 16$ cells), early TP (sky blue, $n = 41$ cells), and late TP (navy blue, $n = 56$ cells). Scatter plots from all TPs are also shown (**C,E**). Red dotted lines represent Io = 0.7, IIa = 0.23/0.33, and Iga = 0.4. In addition, 50% (thick line) and 95% (thin line) probability ellipses are also shown. Note that these scatter plots calculated from fixed MCCs look similar to scatter plots calculated from live-cell imaging analysis (shown in Fig. 2F,G). (**F**) Spinning disk confocal microscopy of the BB-array in MCCs of fixed MTECs, prepared from transgenic mice expressing GFP-centrin2 (green) and mRuby3-Cep128 (magenta) at the initial, early, and late TPs. The differentiation processes of the BB-array in mouse tracheal MCCs were classified into five stages based on the values of Io, IIa, and Iga. Unidirectional BB-orientations are shown (arrows). Insets show 7.5-fold magnified images of fluorescent foci. White arrows indicate the BB-orientation of each BB. Orange dotted lines represent the alignments of adjacent BBs. Bars, 5 μm and 500 nm. (**G**) Schematic diagrams of the developmental stages of the BB-array in mouse tracheal MCCs. Source data are available online for this figure.

Quantitative analysis of the deviation of lattice size at the apical surface of MCCs (Appendix Fig. S3), which showed the evenness of lattice structures, suggested that lattice-like networks of keratin8 gradually developed into ordered structures in mature MCCs, although the lattice-like networks of apical MTs were constant from early to late TPs (Fig. 5F). Since we observed both a decrease in the fluctuation of BB-BF angles and an increase in the BB-global alignment at late TPs using live-cell imaging, we assumed that the lattice-like patterning of apical IFs was a suitable candidate as the regulator restricting the BB-BF angle movement and controlling BB-global alignment in mature tracheal MCCs.

## BB-orientation and BB-alignment in mature MCCs at the late TP were more resistant to nocodazole treatment compared to immature MCCs at the early TP

To elucidate the contribution of the apical cytoskeleton to regular BB-arrays in murine tracheal MCCs experimentally, we conducted MT and IF perturbation experiments in MTECs. First, to investigate the roles of apical MTs in BB-arrays, we treated MTECs with nocodazole at the initial, early, and late TPs (Figs. 6, EV3 and Appendix Fig. S4). Long-term treatment with nocodazole has been reported to affect distribution of PCP proteins especially in immature MCCs (Vladar et al, 2012). We found that a 2 h treatment of MTECs with 6.6 μM nocodazole did not significantly affect the distribution of PCP proteins and instead only disrupted MTs (Appendix Fig. S4).

Nocodazole treatment destroyed the apical networks of both MTs and keratin8 in MCCs of MTECs at initial and early TPs, and BBs showed a scattered or floret-like localization pattern (Figs. 6A–C, and EV3A), consistent with our previous findings (Herawati et al, 2016). However, nocodazole treatment did not destroy the lattice-like network of apical keratin8 in mature MCCs of MTECs at the late TP; we could not find the floret-like localization pattern of BBs, and stages 4–5 MCCs were still retained (Figs. 6A–C and EV3). Notably, BBs were surrounded by keratin8 networks in these MCCs, suggesting that the well-organized lattice-like IFs at the late TP may be stable without dynamic MT networks, and might retain proper BB-arrays at least for a period as short as that in our experimental setup. Quantitative analysis of the BB-array confirmed these observations (Figs. 6D,E and EV3B–D). Notably, there were no significant differences in the Io/IIa/Iga between mature MCCs with or without nocodazole treatment (Figs. 6D,E and EV3D). While we do not dismiss the role of lattice-like MT networks in late TPs, our findings strongly support the occurrence of a spatiotemporal change of major functional apical cytoskeleton networks during MCC differentiation.

## Keratin8-knockout (KO) experiments revealed that the lattice-like networks of apical IFs were required for BB-global alignment and efficient mucociliary clearance

Next, to confirm the contribution of the lattices of apical IFs to the formation and/or stabilization of the regular BB-array in tracheal MCCs, we examined tracheal MCCs in keratin8-KO mice (Fig. 7A). Keratin8 is reported to be expressed in basal luminal progenitor cells during the airway epithelial differentiation in both development and regeneration (Rock et al, 2009; Mori et al, 2015; Watson et al, 2015; Ruiz Garcia et al, 2019). Keratin8-KO mice were reported to show high embryonic lethality in a background strain-dependent manner (Baribault et al, 1994; Ameen et al, 2001). The epithelial cells in the keratin8-KO mice, which survived to adulthood, showed the tissue-dependent IF formation by related keratins such as keratin7 and keratin19 (Ameen et al, 2001; Oriolo et al, 2007a). Therefore, we confirmed the disrupted localization of the apical IF network of MCCs in our keratin8-KO mice by immunofluorescence microscopy using anti-Odf2 and anti-keratin19 antibodies, which stained transition fibers around BBs (Fig. 1C) and apical IFs (Fig. 7B), respectively. Notably, we found that in tracheal MCCs of our keratin8-KO mice, keratin19-positive apical IFs were highly disrupted compared to those in wild-type mice (Fig. 7B). In addition, while no morphological defects were found in the lung tissues of keratin8-KO mice, keratin19-positive IFs were still present (Appendix Fig. S5). This observation aligns with the previously proposed idea of tissue-dependent compensatory IF formation.

While mutations of epithelial keratins have been reported to be related to the defects in liver, intestines, and lungs (Strnad et al, 2008; Polari et al, 2020; Kim et al, 2021), knockout phenotypes in tracheal cells have not yet been explored. To elucidate the contribution of apical IFs in mucociliary clearance, we examined the mucociliary clearance phenotypes of keratin8-KO mice by live analyses of fluorescent beads on tracheal tissues prepared from wild-type and keratin8-KO mice (Figs. 7C,D and EV4A, Movies EV10 and 11). We found that fluorescent beads uniformly flowed in a lung-to-mouth direction on the tracheal surface in wild-type mice. By contrast, movement of the fluorescent beads on the tracheal surface in keratin8-KO mice was severely disturbed.

To elucidate the roles of the BB-array in impaired mucociliary clearance in the tracheas of keratin8-KO mice, we investigated the indexes of BB-alignment and orientation in tracheal MCCs prepared from wild-type and keratin8-KO mice using immunofluorescence microscopy (Fig. 7E–I). BFs were stained using centriolin, as shown in Fig. 1D. BBs were immunofluorescently stained using Odf2, which localized in the anchoring fibers of BBs

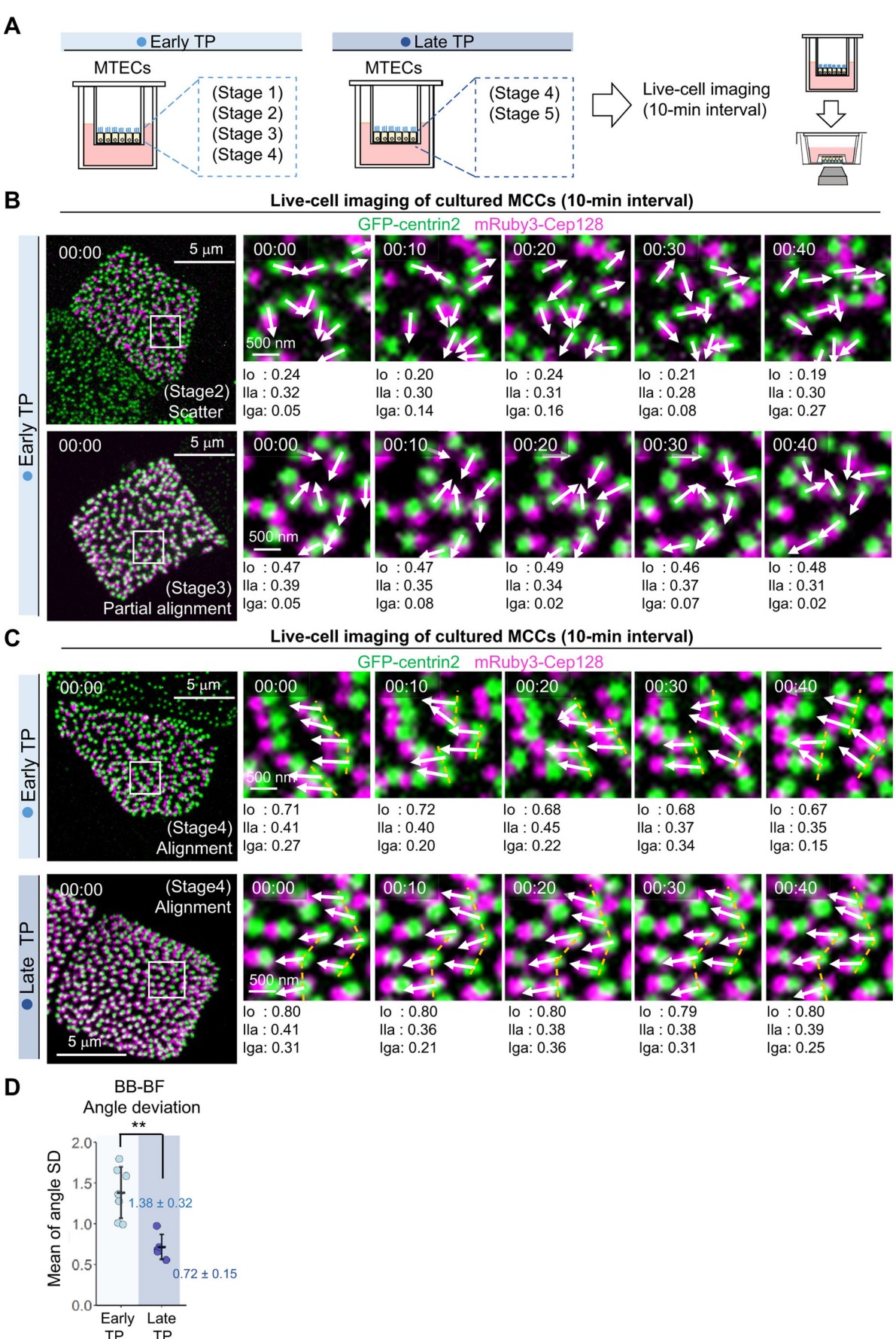

**Figure 4.  Short-interval live-cell imaging analysis of MCCs showing restricted fluctuations in BB-BF angles at late TPs.**

(A) Schematic diagrams of the procedure for short-interval live-cell imaging analysis. MTECs prepared from transgenic mice expressing GFP-centrin2 and mRuby3-Cep128 were observed at early and late TPs with 10-min intervals. MTECs at the early TP included stages 1–4 MCCs, and MTECs at the late TP included stages 4–5 MCCs. (B) Short-interval live-cell imaging analysis of MTECs prepared from transgenic mice expressing GFP-centrin2 (green) and mRuby3-Cep128 (magenta) at the early TP with 10-min intervals. Stage 2 (upper panels) and stage 3 (lower panels) cells are shown. The right panels show 4.5-fold magnified images of fluorescent foci. Time is denoted in hours: minutes. White arrows indicate the BB-orientations of each BB. Bars, 5 μm and 500 nm. See also Movies EV6–7. (C) Short-interval live-cell imaging analysis of MTECs prepared from transgenic mice expressing GFP-centrin2 (green) and mRuby3-Cep128 (magenta) at early/late TPs with 10-min intervals. A stage 4 cell at the early TP (upper panels) and a stage 4 cell at the late TP (lower panels) are shown. Right panels show 4.5-/3.9-fold magnified images of fluorescent foci. Time is denoted in hours: minutes. White arrows indicate the BB-orientations of each BB. Orange dotted lines represent the alignments of adjacent BBs. Bars, 5 μm and 500 nm. See also Movies EV8–9. Note that the MCC at the late TP showed significantly restricted BB movement and BB-BF angle fluctuation. (D) Dot plots of the BB-BF angle deviation in MCCs from MTECs at early ($n = 7$ cells) and late ($n = 5$ cells) TPs observed in our short-interval live-cell imaging analysis. Values are means of each BB-BF angle deviation in individual cells ± standard deviations. **$P < 0.01$ (one-tailed Welch's $t$-test). Source data are available online for this figure.

and in the BFs, thus showing the position of BBs based on the BB-encircling patterns of immunofluorescence (Fig. 7E) (Lange and Gull, 1995; Nakagawa et al, 2001; Ishikawa et al, 2005; Kunimoto et al, 2012; Tateishi et al, 2013; Kashihara et al, 2019). This observation was in line with results from previous work (Herawati et al, 2016). Subsequent immunofluorescence microscopy demonstrated that tracheal tissues prepared from wild-type and keratin8-KO mice both showed a unidirectional BB-orientation, although MCCs in the tracheal tissues from keratin8-KO mice frequently showed abnormal BB-global alignment (Fig. 7F). Quantification of immunofluorescence micrograph data revealed the disrupted BB-global alignment in tracheal MCCs prepared from keratin8-KO mice (Figs. 7G–I and EV4B,C). In tracheal MCCs from wild-type mice, 75.6% of cells were classified as stage 5. However, only 31.1% of cells were classified as stage 5 in tracheal MCCs prepared from keratin8-KO mice; 42.2% of MCCs remained in stage 4, whereas 26.7% of MCCs were classified as stages 2 and 3, suggesting that apical IF lattices were required for the formation and/or stabilization of the final step of the developmental process of murine tracheal MCCs. These results strongly supported the hypothesis that the increase in the lattice-like structure of apical IFs contributed to the BB-global alignment in tracheal MCCs and highlighted the importance of the apical IFs lattice and MTs for efficient mucociliary clearance.

## Keratin8-KO experiments combined with nocodazole treatment revealed the roles of apical cytoskeleton networks in BB-array formation

While we found that apical IF lattices were required for the BB-global alignment, keratin8-KO mice also showed decreased BB-orientation and BB-local alignment which were established at early TPs in MTECs (Fig. EV4). Moreover, 31.1% of cells were still classified as stage 5 in tracheal MCCs of keratin8-KO mice (Fig. 7). These results suggested a more intricate crosstalk between apical cytoskeleton network and BBs. For detailed analysis, we performed a nocodazole-treatment assay using tracheal MCCs from the wild-type and keratin8-KO mice (Fig. 8). In this experiment, we examined keratin8-KO mice expressing GFP-centrin2 and mRuby3-Cep128, anticipating similar observation conditions as in the experiments using MTECs (Fig. 8A).

To determine the effects of nocodazole treatment on BB-arrays in tracheal MCCs, we confirmed that a 2 h treatment of the trachea with 9.9 μM nocodazole effectively disrupted MT networks in tracheal MCCs (Figs. 8B and EV5A). We also confirmed that, when incubated with DMSO, keratin8-KO mice expressing GFP-centrin2

and mRuby3-Cep128 exhibited phenotypes similar to the keratin-KO mice shown in Fig. 7 (Figs. 8C and EV5B). Interestingly, in nocodazole-treated wild-type tracheal MCCs (Figs. 8D and EV5C), we did not observe any destruction of the BB-array, a result similar to that in MTECs (Fig. 6). This supported the idea that IF-lattice could maintain an established BB-array without MT networks in the short-term. In contrast, in nocodazole-treated keratin8-KO tracheal MCCs, we observed a highly disturbed BB-array (Figs. 8D and EV5C). Quantification of immunofluorescence micrographs revealed that the number of stage 5 MCCs was decreased by the nocodazole treatment in keratin8-KO tracheal MCCs (Fig. 8C,D). Furthermore, BB-orientation and BB-global alignment were significantly diminished (Figs. 8E,F and EV5C,D). These findings suggest that a well-organized lattice-like structure composed of IFs surrounding BBs was required to increase the BB-global alignment and maintain high levels of BB-orientation and BB-local/global alignment in mature MCCs. The apical MTs, crucial for establishing BB-orientation and local BB-alignment in immature MCCs, also contribute to BB-orientation and BB-global alignment in mature MCCs to some extent.

## Discussion

In the current study, we used dual-color live-cell imaging of BBs and BFs to assess the real-time relationships among BB-orientation, BB-local alignment, and BB-global alignment. In previous single-color live-cell imaging studies of BBs, BB-local alignment was analyzed in real time, and its relationship with BB-orientation was indirectly discussed from fixed images. Accordingly, in this study, we used mathematical quantification and pharmacological and genetic interventions and showed that total coordination of unidirectional BB-orientation and BB-global alignment was achieved during the final MCC differentiation. This coordination relies on the spatiotemporal interplay of the apical cytoskeleton networks of MTs and IFs (Fig. 9).

Unidirectionally coordinated BB-orientation is absolutely required for coordinated multiciliary beating in MCCs (Brooks and Wallingford, 2014) in various organs, including the trachea (Vladar et al, 2012), oviducts (Usami et al, 2021), and brain ventricles (Guirao et al, 2010), because cilia originate from ciliary BBs and the BB-orientation determines the direction of the cilia beating (Kunimoto et al, 2012; Clare et al, 2014). By contrast, the positional distributions of BBs have not been studied in detail, although some reports have shown that the overall BB distribution differs among cell types. In ependymal MCCs, BBs cluster in an asymmetrical distribution in the apical plane (Boutin

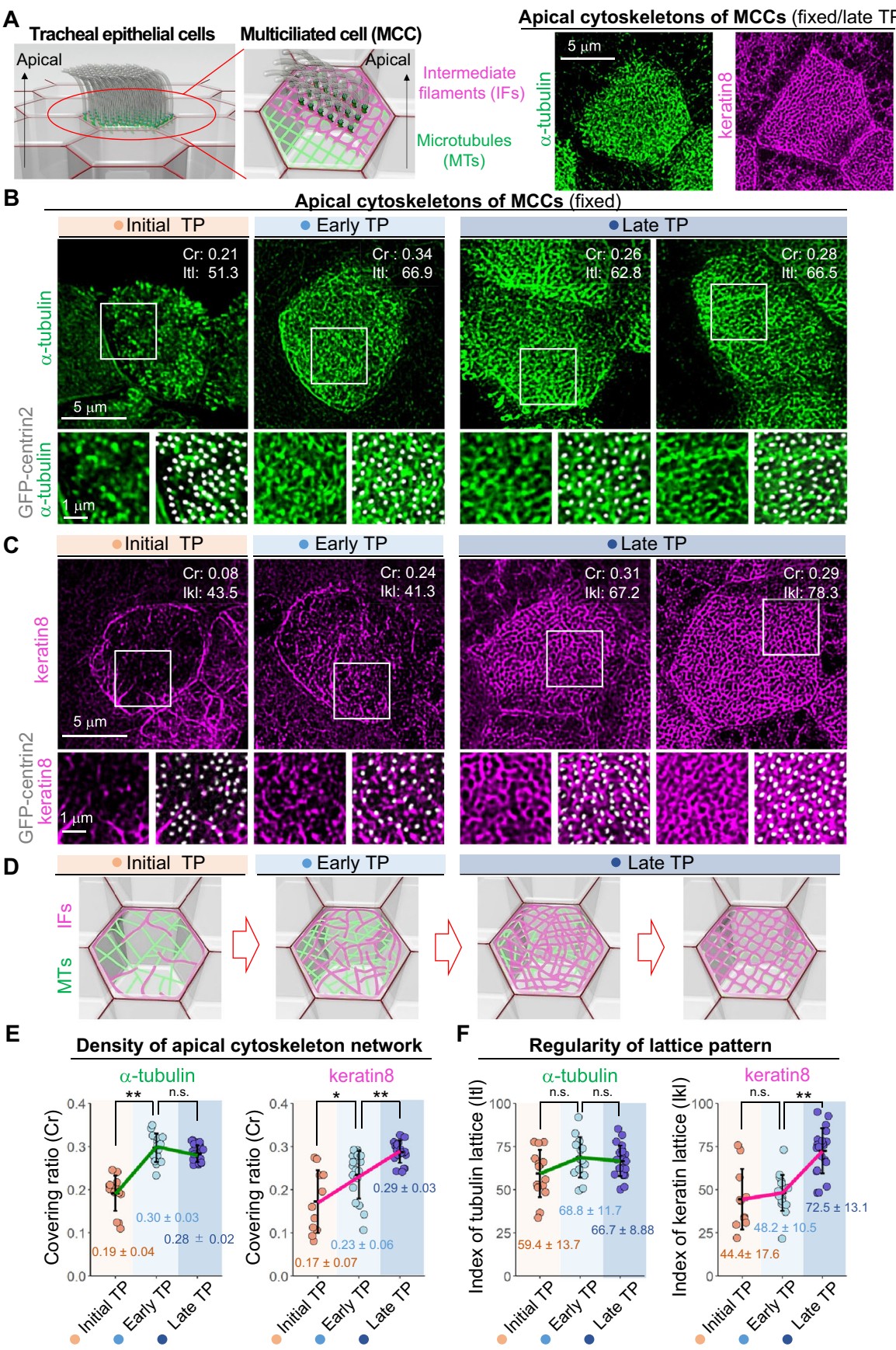

**Figure 5. Developmental process of apical cytoskeleton networks during MCC differentiation.**

(A) Schematic diagrams of apical cytoskeleton networks in MCCs of MTECs (left panels) and spinning disk confocal microscopy of fixed MTECs prepared from transgenic mice expressing GFP-centrin2 and mRuby3-Cep128 at the late TP. MCCs of MTECs were fixed and stained using anti-α-tubulin antibodies (green) or anti-keratin8 antibodies (magenta). Bar, 5 μm. (B,C) Spinning disk confocal microscopy of fixed MTECs prepared from transgenic mice expressing GFP-centrin2 and mRuby3-Cep128 at initial, early, and late TPs. MCCs of MTECs were fixed and stained using anti-α-tubulin antibodies (green in (B)) or anti-keratin8 antibodies (magenta in (C)). Insets show 1.8-fold magnified images of fluorescent foci. Bars, 5 and 1 μm. The covering rate (Cr) and the indexes of the tubulin lattice (Itl) and keratin lattice (Ikl) are shown. See also Materials and Methods. (D) Schematic diagrams of the development of the apical cytoskeleton in MCCs of MTECs. Apical MTs (green) and keratin (magenta) networks are shown. (E) Dot plots of the Cr of the apical cytoskeleton of α-tubulin and keratin8 in MCCs at initial (pink, $n = 15$ cells and $n = 11$ cells), early (sky blue, $n = 15$ cells and $n = 18$ cells), and late (navy blue, $n = 23$ cells and $n = 21$ cells) TPs. Green/magenta lines represent the mean for each TP. Values are means ± standard deviations (SDs) of each index. *$P < 0.05$, **$P < 0.01$, n.s., not significant (Kruskal–Wallis and pairwise-comparison test). (F) Dot plots of the index of regularity for the lattice-like patterns of α-tubulin (Itl) and keratin8 (Ikl) in MCCs of MTECs at initial (pink, $n = 15$ cells and $n = 11$ cells), early (sky blue, $n = 15$ cells and $n = 18$ cells), and late (navy blue, $n = 23$ cells and $n = 21$ cells) TPs. Green/magenta lines represent the mean of each TP. Values are the means ± SDs of each index. **$P < 0.01$, n.s., not significant (one-way analysis of variance with Tukey-Kramer multiple-comparison test). Source data are available online for this figure.

et al, 2014). In *Xenopus* larval skin MCCs, BBs are evenly distributed over their apical planes (Werner et al, 2011). In addition, in mammalian tracheal and oviductal MCCs, BBs have been reported to be aligned (Kunimoto et al, 2012; Herawati et al, 2016; Tateishi et al, 2017; Usami et al, 2021). Here, by live imaging of each step of the differentiation of BB-arrays, we demonstrated the apparent biphasic development of BB-orientation and BB-local and -global alignment.

Cytoskeletons play multifaceted roles in cells, including roles in cell motility, cell polarization, signal transduction, cell fate switching, and cell differentiation (Janmey, 1998; Mammoto and Ingber, 2009; Fletcher and Mullins, 2010). With respect to epithelial differentiation, polarized MT arrays oriented along the apicobasal axis have been reported to regulate intestinal apicobasal polarity (Toya et al, 2016; Noordstra et al, 2016). Cortical actin filaments are important for the emergence and expansion of the apical surface and ciliogenesis in *Xenopus* larval skin MCCs (Werner et al, 2011). Regarding IFs, keratin filaments have been shown to reinforce mechanical strength by coordinating with desmosomes (Hatzfeld et al, 2017) and stabilize the apicobasal polarity of epithelial cells (Oriolo et al, 2007b). In addition to these roles of the cytoskeleton in epithelial differentiation, unique cytoskeletons composed by MTs, actin, and IFs were identified in mammalian tracheal MCCs almost four decades ago, albeit without functional analyses or definition of their unique specific structures (Gordon, 1982). Recently, the cytoskeletons were shown to form a characteristic apical layered structure and to exist in epithelial cells in general (Tateishi et al, 2017; Yano et al, 2017). Thus, in the current study, we demonstrated the specific roles of apical MTs and apical IFs in BB-orientation, BB-local alignment, and BB-global alignment during MCC development. In this study, we found that in fully differentiated tracheal MCCs, lattice-like apical IFs stabilized BB-global alignment, formed stage 5 MCCs, and also suppressed the fluctuations in BBs, likely leading to the coordination of ciliary beating. Accordingly, our work may provide insights into the novel roles of apical IFs and apical MTs in tracheal MCCs to facilitate efficient mucociliary transport.

Understanding how apical cytoskeleton networks and BB-arrays are coordinated in the tracheal MCCs remains challenging. BF is a strong candidate that functions as a bridge between BBs and apical cytoskeletons, because it contains MT-binding proteins such as γ-tubulin ring complex (γ-TuRC), NEDD1, Cep170, and ninein and is thought to function as an MTOC (Nguyen et al, 2020). Consistently, loss of BF in the Odf2 mutant mice caused a significant reduction in apical MT networks (Kunimoto et al, 2012; Tateishi et al, 2017). Our findings, which indicate the importance of

MT networks for BB-orientation in immature/mature tracheal MCCs (Figs. 6 and 8), align these studies. Our current work also suggests that dense lattice-like IF network formation surrounding BBs strongly correlates with an increase in the BB-global alignment in mature tracheal MCCs. While the direct interaction between BB and IF in MCCs is not clear, some BF components were reported as binding partners of IFs. For instance, trichoplein (TCPH)/mitostatin, a keratin IF scaffold protein, has been reported to interact with Odf2, playing a role in the recruitment of ninein (Ibi et al, 2011). Furthermore, GCP6, a component of γ-TuRC, was reported to interact with IFs in intestinal epithelial cells (Oriolo et al, 2007a). Given that MT networks also contribute to BB-global alignment to some extent (Fig. 8), it is plausible that BF or BF-related MT networks including IF binding proteins could be instrumental regulators of IF network formation.

BB-local alignment increased in an MT-dependent manner in immature MCCs but was not affected by nocodazole treatment in mature MCCs (Figs. 6 and 8). This implies that other structures might play a role in stabilizing the shorter lines of BB in mature BBs. Beyond MTs and IFs, subapical actin filaments in Xenopus epidermal MCCs have been reported to connect BFs to neighboring ciliary rootlets (Werner et al, 2011; Antoniades et al, 2014). Such structures, however, were not identified in mouse tracheal MCCs (Tateishi et al, 2017), suggesting the need for further investigation.

PCP signaling, a conserved signaling pathway in the animal epithelium, is a main regulator of BB-orientation in MCCs and is achieved via apical MTs (Oshita et al, 2003; Guirao et al, 2010; Vladar et al, 2012; Boutin et al, 2014; Nakayama et al, 2021; Sone et al, 2021). In our current study, we did not observe the direct effects of PCP signaling to IFs. However, in terms of the sequential organization process, apical MTs were formed prior to the formation of IFs. We also demonstrated that nocodazole treatment of immature MCCs disrupted IF localization (Herawati et al, 2016). In addition, loss of MTs in ODF2 mutants also caused abnormal IF bundling (Kunimoto et al, 2012; Tateishi et al, 2017). These results suggested that pre-existing MT networks regulated by PCP signaling may function as scaffolds for subsequent IF formation. Interestingly, the localization of plectin (Svitkina et al, 1996), a major IF-based crosslinker protein among MTs, IFs, and actin was found in apical cytoskeletons of MCCs (Herawati et al, 2016). Further studies of the direct relationships between PCP signaling and apical IFs as well as crosstalk among multiple cytoskeletons may provide insights into the organization process of the unique apical cytoskeletal structures of tracheal MCCs.

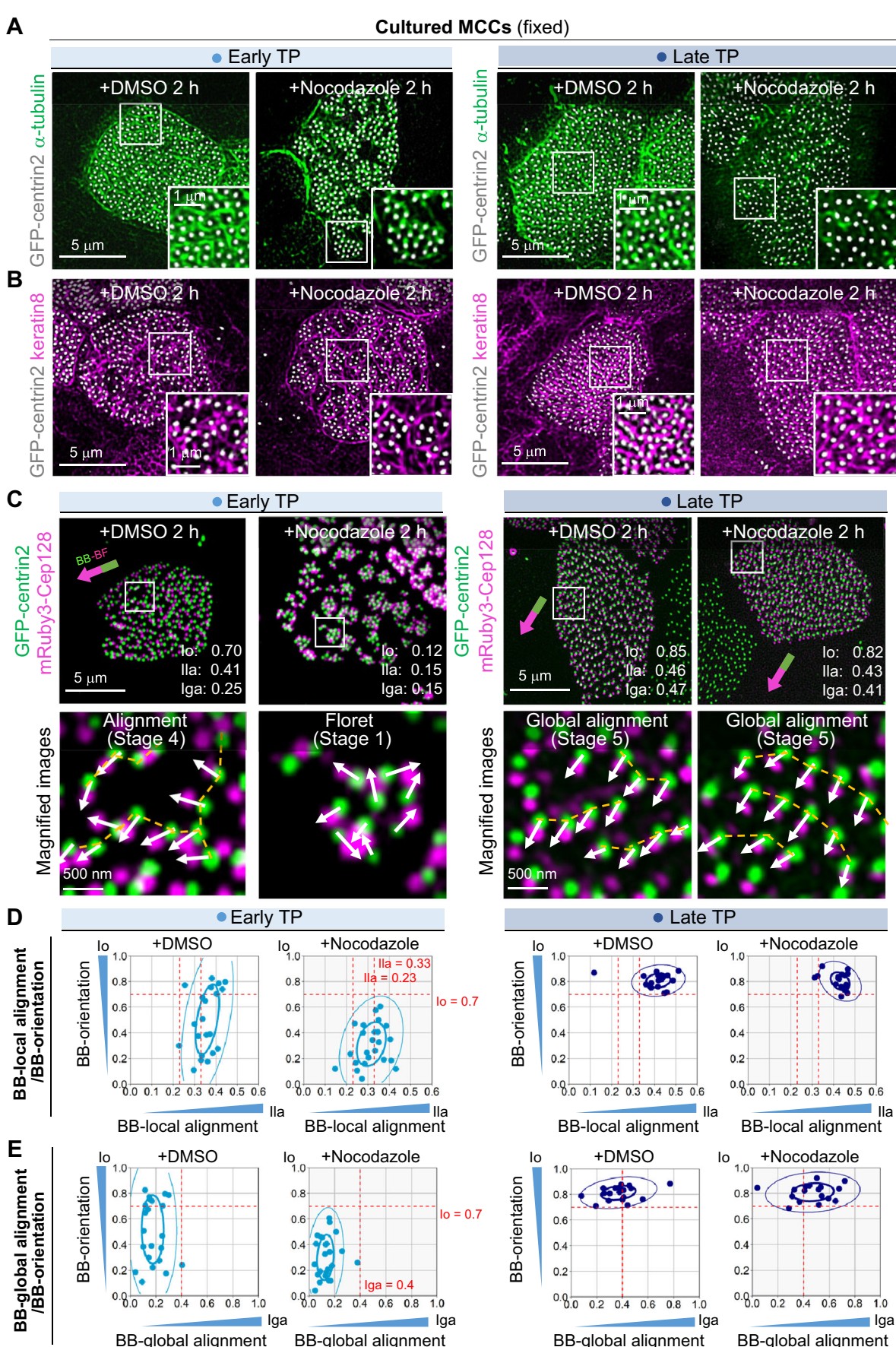

**Figure 6. Quantitative analysis of the effects of nocodazole treatment on the BB-array in MCCs of MTECs at different TPs.**

(A,B) Spinning disk confocal microscopy of the apical cytoskeletons in MCCs of fixed MTECs prepared from transgenic mice expressing GFP-centrin2 (gray) and mRuby3-Cep128 at early and late TPs after treatment with DMSO or 6.6 μM nocodazole for 2 h. MCCs of MTECs were fixed and stained using anti-α-tubulin (green in (A)) or anti-keratin8 (magenta in (B)) antibodies. Insets show 2-fold magnified images of fluorescent foci. Bars, 5 and 1 μm. (C) Spinning disk confocal microscopy of the BB-array in MCCs of fixed MTECs prepared from transgenic mice expressing GFP-centrin2 (green) and mRuby3-Cep128 (magenta) at early and late TPs after treatment with DMSO or 6.6 μM nocodazole for 2 h. Insets show 6.7-fold magnified images of fluorescent foci. Bars, 5 μm and 500 nm. (D,E) Scatter plots of Io/IIa (D) and Io/Iga (E) in MCCs of MTECs prepared from transgenic mice expressing GFP-centrin2 and mRuby3-Cep128. The cells were treated for 2 h with DMSO or 6.6 μM nocodazole. Samples at early (sky blue, $n = 22$ cells and $n = 24$ cells) and late (navy blue, $n = 18$ cells and $n = 17$ cells) TPs are shown. Red dotted lines represent Io = 0.7, IIa = 0.23/0.33, and Iga = 0.4. In addition, 50% (thick line) and 95% (thin line) probability ellipses are shown. Note that the nocodazole retreatment did not affect the BB-arrays at the late TP. Source data are available online for this figure.

In addition to mucociliary clearance, which acts as a physical barrier, tracheal epithelial cells adhere to each other and provide a tight junction (TJ)-based paracellular barrier (Anderson and Van Itallie, 2009; Yano et al, 2017). The IFs and MTs were positioned under the membrane around the BF level (50–200 nm below the apical surface) (Tateishi et al, 2017), consistent with the aspect that TJs anchor the apical cytoskeleton because TJs are localized at the most apical parts of the lateral membranes of epithelial cells (Tateishi et al, 2017; Yano et al, 2017; Yano et al, 2021). These data made us wonder whether the stepwise organization of apical cytoskeletons was related to the epithelial barrier function during MCC differentiation. Intriguingly, analysis of transepithelial electrical resistance (TER) for evaluation of epithelial barrier function demonstrated that TERs were stabilized in MTECs after ALI14 (Davidson et al, 2000; You et al, 2002), equivalent to our mature MTECs. Therefore, further studies are needed to assess the crosstalk between multiple apical cytoskeletons and the associations of apical cytoskeleton networks with cell-cell adhesion sites to improve our understanding of the physical barriers of tracheal epithelial cells.

Further analysis of the regulators of multiple cytoskeleton networks, which may be related to PCP, as well as PCP effector proteins and protein functions in cell-cell adhesions, may provide insights into the general mechanisms that coordinate BB-arrays and unidirectional ciliary beating in mammalian tracheal tissues. Mucociliary clearance is an important self-defense mechanism established by tracheal epithelial cells and includes mucus and liquid secretion by secretory cells and coordinated ciliary beating by MCCs (Tilley et al, 2015; Legendre et al, 2021). Therefore, understanding the regulatory factors in coordinated ciliary beating is becoming increasingly important. This study will improve our understanding of the regulation of coordinated ciliary beating for mucociliary clearance and the pathophysiology of related airway diseases for medical innovation.

## Methods

### Animals

All experimental and housing conditions of animals were approved by the Institutional Animal Care and Use Committee of Osaka University (FBS-17-003). The mice used in the current study included wild-type C57BL/6J mice (CLEA Japan, Inc., Tokyo, Japan, RRID:IMSR_JCL:MIN-0003), GFP-centrin2 transgenic mice (Higginbotham et al, 2004), mRuby3-Cep128 transgenic mice (generated as described below), and keratin8-KO mice (cat. no. 007031; The Jackson Laboratory, Farmington, CT, USA, RRID:IMSR_JAX:007031). To generate GFP-centrin2; mRuby3-Cep128 mice, homozygous GFP-centrin2 mice were crossed with homozygous mRuby3-Cep128 mice. Homozygous GFP-centrin2; mRuby3-Cep128 mice were obtained by heterozygous mating.

### Generation of mRuby3-Cep128 transgenic mice

The mRuby3-Cep128 transgene was excised from the backbone vector using SspI and AvrII restriction enzymes, separated by agarose gel electrophoresis, purified with a Qiagen gel extraction kit (Qiagen, Valencia, CA, USA), and then injected into one-cell embryos of C57BL/6J mice.

### Primary culture of MTECs

Primary cultures of MTECs were prepared from the tracheas of mice at 7–20 weeks of age, as described previously (You et al, 2002), with modifications (Herawati et al, 2016; Nakayama et al, 2021). The tracheas of mice were dissected from the larynx to the bronchial first branches and collected in Ham's F-12 medium with penicillin-streptomycin at 4 °C. The connective tissue was removed from the tracheas in cold medium (4 °C) under stereomicroscope observation. Murine tracheas were washed with cold medium and incubated in Ham's F-12 with penicillin-streptomycin containing 1.5 mg/mL pronase (Roche Molecular Biochemicals, Indianapolis, IN, USA) at 4 °C for 16–20 h with gentle rocking for enzymatic digestion after the longitudinal dissection of tracheas. The digestive reaction was stopped by adding fetal bovine serum (FBS; cat. no. 172012; Sigma-Aldrich, St. Louis, MO, USA) to the tube for enzymatic digestion of the trachea to a final concentration of 10%. The solution tube containing MTECs was centrifuged at $400 \times g$ for 10 min at 4 °C, and pellets of MTECs were resuspended in MTEC Basic Medium (You et al, 2002) with 10% FBS for 3.5 h at 37 °C in an atmosphere containing 5% $CO_2$ to exclude contaminating fibroblasts via attachment to the culture dish. Nonadherent MTECs were collected by centrifugation, counted, resuspended in BEGM medium (cat. no. CC-3170; Lonza, Basel, Switzerland), and seeded at $10^5$ cells/cm² onto polyester 24-well transwell filters (cat. no. 3470; Corning, NY, USA) coated with 50 μg/mL collagen type I (cat. no. 354249; Corning). MTEC cultures were maintained in the proliferation stage using BEGM medium. After confirmation of the confluent cell sheet under bright-field observation (after 4–6 days of culture), air-lift of the cell sheet was conducted, and the medium was changed to Pneumacult-ALI (cat. no. 05001; STEMCELL Technologies, Vancouver, Canada).

### Immunofluorescence analysis

MTECs and whole-mount tracheas were fixed with ice-cold methanol at –20 °C for 10 min or with 4% paraformaldehyde in HBS at room temperature for 15 min, washed with phosphate-buffered saline (PBS)

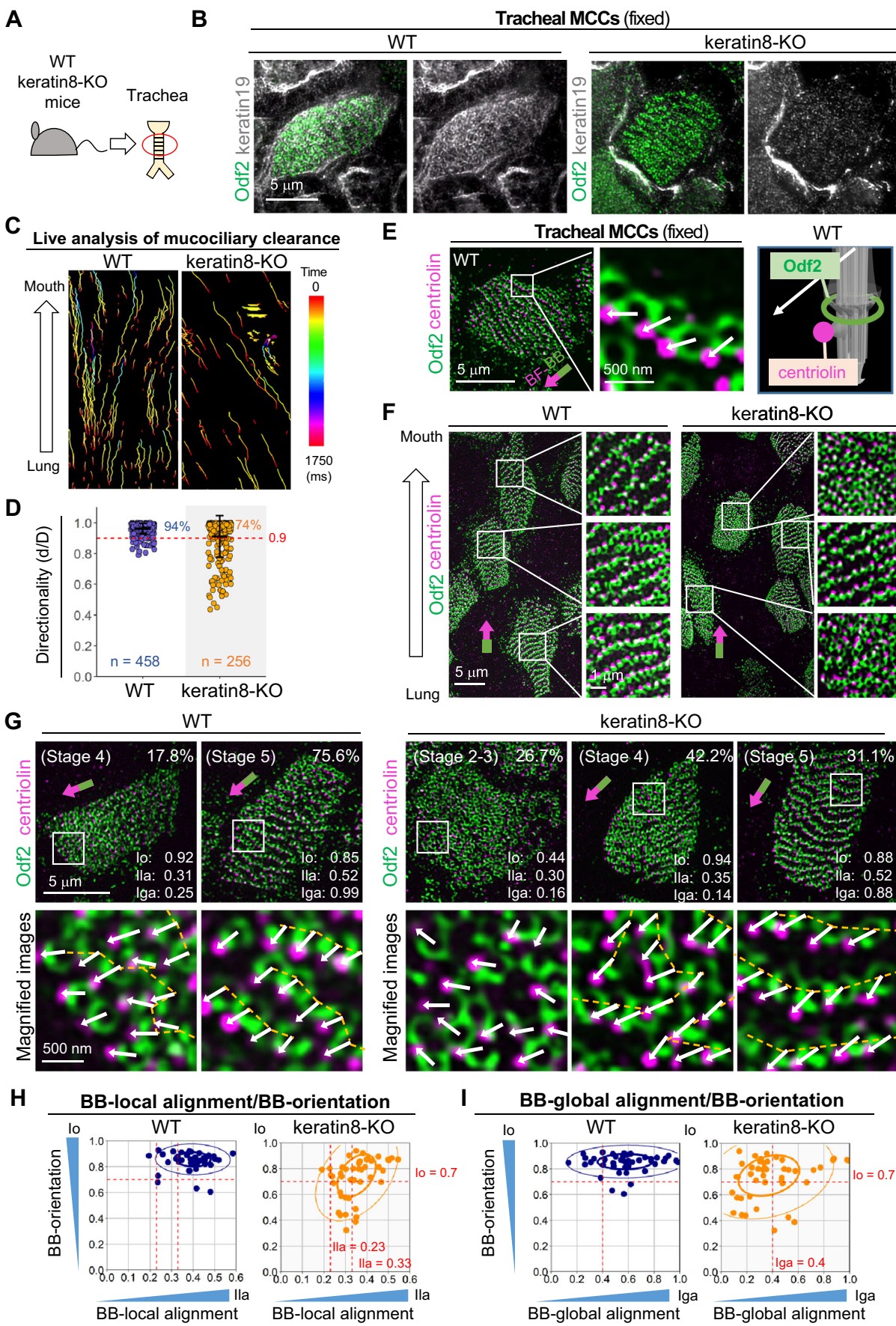

**Figure 7. Keratin8-knockout experiments of mucociliary clearance and quantitative analysis of the BB-array in MCCs.**

(A) Schematic diagrams of the procedure for the quantitative analysis using keratin8-KO mice. (B) Spinning disk confocal microscopy of MCCs of fixed tracheal cells prepared from wild-type and keratin8-KO mice. MCCs of tracheal cells were fixed and stained using anti-Odf2 (green) and anti-keratin19 (gray) antibodies. Bar, 5 μm. (C) Mucociliary clearance analysis in isolated tracheal tissues of wild-type and keratin8-KO mice using live imaging of fluorescent beads. The positions of the beads at each timepoint are indicated by a color corresponding to the time scale bar (0–1750 ms) in the right panel. Each bead was adjusted to a uniform starting time. Whole images are shown in Fig. EV4A. (D) The directionality of the mucociliary clearance of the beads was estimated by the ratios of end-to-end distances (d) against distances along the bead trajectory (D) of the fluorescent beads at the isolated tracheal surfaces from wild-type (navy blue, $n = 458$) and keratin8-KO (orange, $n = 256$) mice. The red dotted line represents d/D = 0.9. Note that 94% of the beads in wild-type trachea and 74% of the beads in keratin8-KO trachea uniformly flowed in our assay (0.9 ≤ d/D). (E) Spinning disk confocal microscopy of the BB-array in MCCs of fixed tracheal cells prepared from wild-type and keratin8-KO mice. MCCs of tracheal cells were fixed and stained using anti-Odf2 (green) and anti-centriolin (magenta) antibodies. Bars, 5 μm and 500 nm. Schematic diagrams of a cilium/BB/BF complex in mouse tracheal MCCs are shown (right panel). White arrows indicate the BB-orientation of each BB. Bars, 5 μm and 500 nm. Note that the ring-like structures are BBs stained by anti-Odf2 antibodies. (F) Spinning disk confocal microscopy of the BB-array in MCCs of fixed tracheal cells prepared from wild-type and keratin8-KO mice. MCCs of tracheal cells were fixed and stained using anti-Odf2 (green) and anti-centriolin (magenta) antibodies. Bars, 5 μm. Insets show 3.6-fold magnified images of fluorescent foci. (G) Spinning disk confocal microscopy of the BB-array in MCCs of fixed tracheal cells prepared from wild-type and keratin8-KO mice. MCCs of tracheal cells were fixed and stained using anti-Odf2 (green) and centriolin (magenta) antibodies. Representative cells at each stage are shown. All images in Fig. 7 are derived from the same experiment and wild-type Stage 5 cell was selected from the image shown in (E) and (F), and keratin8-KO Stage 4 and 5 cells were selected from one image shown in F (see also Source Data). Bars, 5 μm. Insets show 6.3-fold magnified images of fluorescent foci. White arrows indicate the BB-orientations. Orange dotted lines represent the alignments of adjacent BBs. (H,I) Scatter plots of Io/IIa (H) and Io/Iga (I) in MCCs of tracheal cells prepared from wild-type (navy blue, $n = 45$ cells) or keratin8-KO (orange, $n = 45$ cells) mice. Red dotted lines represent Io = 0.7, IIa = 0.23/0.33, and Iga = 0.4. In addition, 50% (thick line) and 95% (thin line) probability ellipses are shown. Source data are available online for this figure.

three times, permeabilized with 0.2% Triton X-100 in PBS at room temperature for 15 min, and incubated in 1% bovine serum albumin in PBS at 4 °C for 30 min. The samples were then incubated with primary antibodies at room temperature for 1 h, washed with PBS three times, and incubated with secondary antibodies at room temperature for 1 h. After three washes in PBS and one wash in MilliQ water, the samples were mounted in fluorescent mounting medium (Dako fluorescent mounting medium; Agilent Technologies, Santa Clara, CA, USA). Images were recorded using a Spinning Disk-Olympus super-resolution microscope (SD-OSR; SpinSR; Olympus, Tokyo, Japan) equipped with a silicone oil immersion objective lens (UPlanSApo 60× Sil, NA 1.3; Olympus; UPlanSApo 30× Sil, NA 1.05; Olympus) and an ORCA-Flash 4.0v2 sCMOS camera (Hamamatsu Photonics, Shizuoka, Japan) or a Zyla 4.2 sCMOS camera (Andor Technology, Belfast, UK). Image deconvolution was performed using cellSens Dimension Desktop 3.2 Software (Olympus, RRID:SCR_014551). Images were prepared using Adobe Photoshop, Illustrator (Adobe Systems, San Jose, CA, USA, RRID:SCR_014199, RRID:SCR_010279) and Fiji Image J2 (National Institutes of Health, Bethesda, MD, USA, RRID:SCR_002285). Maximum intensity projections from 25–30 optical planes covering approximately the whole height of the cells or 5–7 optical planes covering approximately the apical regions of the cells (Figs. 5 and 6A,B) were reconstituted using Image J2 Fiji software.

## Antibodies

The following primary antibodies were used in this study: mouse monoclonal antibodies against α-tubulin (cat. no. T9026; Sigma-Aldrich, RRID:AB_477593), Odf2 (Fig. 7B; cat. no. H00004957-M01; Abnova, Taipei, RRID:AB_1137338), centriolin (Fig. 7E–G; cat. no. sc-365521; Santa Cruz Biotechnology, Santa Cruz, CA, USA, RRID:AB_10851483), keratin8 (RL273, Cosmo Bio, Japan), keratin18 (cat. no. C8541; Sigma-Aldrich, RRID:AB_476885); rabbit polyclonal antibodies against Cep128 (produced in our laboratory (Kashihara et al, 2019)), Prickle2 (kindly provided by Dr. T. Otsuka, University of Yamanashi, Yamanashi, Japan); rat monoclonal antibodies against α-tubulin (cat. no. ab6160, Abcam, RRID:AB_305328), Odf2 (Fig. 7E–G; made in our laboratory (Tateishi et al, 2013)), centriolin (Fig. 1; made in our laboratory

(Ishikawa et al, 2005)), keratin8 (TROMA-I; DSHB, Iowa City, IA, USA, RRID:AB_531826), and keratin19 (TROMA-III; DSHB, RRID:AB_2133570). The following secondary antibodies were used: Alexa Fluor 568/647 goat anti-mouse IgG (H + L) (Molecular Probes, Eugene, OR, USA), Alexa Fluor 640 goat anti-rabbit IgG (H + L) (Molecular Probes), and Alexa Fluor 488/568/647 goat anti-rat IgG (H + L) (Molecular Probes).

## Live-cell imaging analysis

Transwell filters with MTEC sheets were cut out from the plastic support very carefully 5–14 days after the start of ALI culture. Then, the filters were inverted onto glass-bottom dishes such that the cell sheet surface was in contact with the glass bottom. Medium was supplied from the additional chamber placed on the back surface of inverted transwell filters with MTEC sheets. The medium was changed very carefully using a syringe to avoid moving the position of the MTEC sheets. The movement of BBs (GFP-centrin2) and BFs (mRuby3-Cep128) was recorded using a Spinning Disk-Olympus super-resolution microscope (SD-OSR; SpinSR; Olympus) equipped with a silicone oil immersion objective lens (UPlanSApo 60× Sil, NA 1.3; Olympus), a motorized scanning deck, and the appropriate filter sets for DAPI/FITC/TRITC. The microscope was connected to an ORCA-Flash 4.0v2 sCMOS camera (Hamamatsu Photonics) or a Zyla 4.2 sCMOS camera (Andor Technology). All hardware was controlled using MetaMorph software (Molecular Devices, San Jose, CA, USA, RRID:SCR_002368). The culture conditions during live-cell imaging were maintained using an incubation chamber (37 °C, 5% $CO_2$, 85% humidity; Tokai Hit, Shizuoka, Japan). The recording was performed using a multistage acquisition system to acquire 10–40 fields of view, each covering ~20 cells/field. Time intervals were set as 45 or 60 min for long-term recording and 10 min for short-term recording.

## Quantitative analysis of BB-orientation and BB-alignment

### *Index of orientation (Io)*

To quantify variations in BB-orientation within a single cell, we used the Io, which was used in our previous work (Herawati et al, 2016). Io was calculated as follows. We found the XY coordinates of

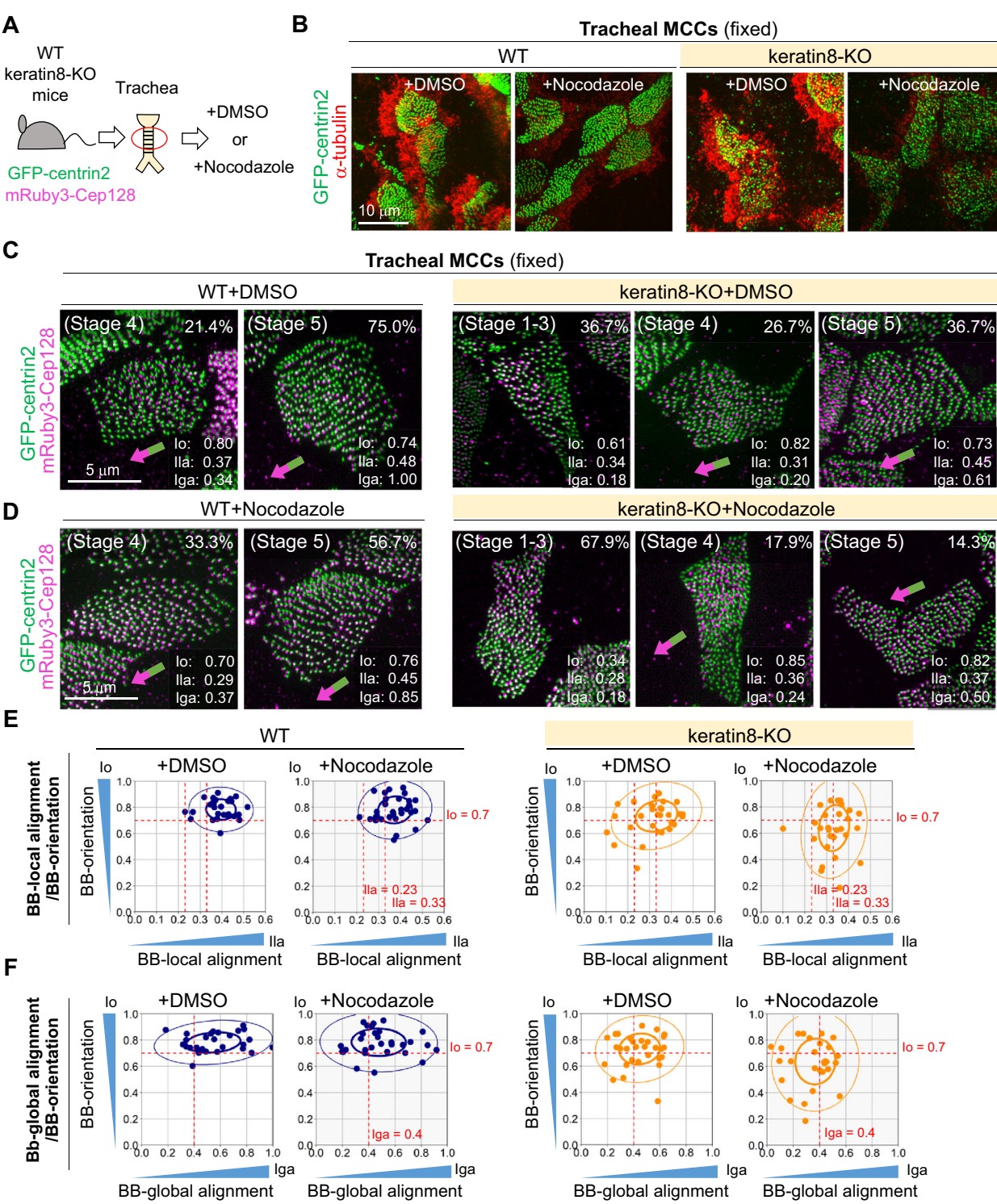

BBs and BFs in a single cell from our microscopy data, and we set BB/BF vectors, each of which originated from the center of a BB and ended at its accompanying BF. The vectors were then divided by their norm and used as unit vectors Pi ($i = 1, 2, \dots n$).

$$P_i = (\cos\theta_i,\ \sin\theta_i) \tag{1}$$

We defined the Io as the norm of a mean vector of vectors Pi ($i = 1, \dots, n$; $n$ = the total number of BBs) within a single cell.

$$\mathrm{Io} = \left| \frac{1}{n} \sum_{i=1}^{n} Pi \right| \tag{2}$$

**Figure 8. Quantitative analysis of the effects of nocodazole treatment on the BB-array in MCCs of keratin8-knockout mice.**

(A) Schematic diagrams of the procedure for the quantitative analysis of the BB-array using keratin8-KO mice expressing GFP-centrin2 mRuby3-Cep128. (B) Spinning disk confocal microscopy of MCCs of fixed tracheal cells prepared from wild-type and keratin8-KO mice expressing GFP-centrin2 (green) and mRuby3-Cep128. MCCs of the tracheal cells were fixed and stained using anti-α-tubulin antibody (red) after treatment with DMSO or 9.9 μM nocodazole for 2 h. Bar, 5 μm. (C,D) Spinning disk confocal microscopy of the BB-array in MCCs of fixed tracheal cells prepared from wild-type and keratin8-KO mice expressing GFP-centrin2 (green) and mRuby3-Cep128 (magenta). MCCs of tracheal cells were fixed after treatment with DMSO (C) or 9.9 μM nocodazole (D) for 2 h. Representative cells at each stage are shown. All images are derived from the same experiment, and we selected two cells from one image of wild-type cells in (D) (see also Source Data). Bars, 5 μm. (E,F) Scatter plots of Io/Ila (E) and Io/Iga (F) in MCCs of tracheal cells prepared from wild-type (navy blue, $n = 28$ cells and $n = 30$ cells) or keratin8-KO (orange, $n = 30$ cells and $n = 28$ cells) mice after treatment with DMSO or 9.9 μM nocodazole for 2 h. Red dotted lines represent Io = 0.7, Ila = 0.23/0.33, and Iga = 0.4. In addition, 50% (thick line) and 95% (thin line) probability ellipses are shown. Source data are available online for this figure.

Note that Io is a value between 0 and 1. If all BBs-BFs pointed in the same direction in a single cell, the Io was 1. If the BB-orientations were completely random, Io was 0.

### Index of local alignment (Ila) and index of linearity (Ilin)

To quantify the degree of BB-local alignment in a single cell, we used the index of alignment (Ia), as previously described (Herawati et al, 2016), as the Ila in this work. We set d = 5, 7, and 8, depending on the magnification settings used for the microscopy analysis. In addition, to calculate the index of global alignment (Iga), we modified the definition of a neighboring BB group and set the Ilin (Appendix Fig. S2A,B). We found the XY coordinates of BBs in a single cell from our microscopy data and defined the neighboring BBs, which were composed of three BBs, i.e., BBi (a BB of interest), BBj (the BB closest to BBi), and BBk (the BB second closest to BBi). The threshold of the distance between the closest BBs was changed from the variable value (1.3d in Ila) into a fixed value of 600 nm in Ilin/Iga. We then calculated $\cos\theta_i$ from the inner product and norm of vectors ($i = 1, …, n$; $n$ = the total number of BBs; $0 \leq \theta_i < 180$, between the vectors of BBi-BBj and BBi-BBk). The Ilin$_i$ was then defined as follows based on whether the distance between BBi and BBk was less than 600 nm:

$$\text{I) BBi} - \text{BBk distance} > 600 \text{ nm, } \text{Ilin}_i = 0 \qquad (3)$$

$$\text{II) Bi} - \text{BBk distance} \leq 600 \text{ nm, } \text{Ilin}_i = \max(-\cos\theta_i, 0) \qquad (4)$$

Note that Ila is a value between 0 to 1. If all BBs aligned in a straight line ($\theta_i = 180$), Ilin was 1. If the BBs aligned in a V-shape ($0 \leq \theta_i \leq 90$), Ilin was 0 (Appendix Fig. S2B).

### Iga

To quantify the degree of BB-global alignment in a single cell, we defined Iga (Appendix Fig. S2C). We found the XY coordinates of BBs in a single cell from microscopy data and defined the neighboring BBs similar to that used for the calculation of A-Ila. In addition, we calculated the $\sin 2\theta_i$ and $\cos 2\theta_i$ using the additive theorem ($i = 1, …, n$; $n$ = the total number of BBs; $0 \leq \theta_i < 180$), and the unit vector vec$_i$ was defined. Note that $\theta_i$ was doubled ($0 \leq 2\theta_i < 360$) to counteract perpendicularly oriented unit vectors (Appendix Fig. S2D).

$$\text{vec}_i = (\cos 2\theta_i, \ \sin 2\theta_i) \qquad (5)$$

To reduce the contribution from unaligned V-shape neighboring BBs, we calculated the values of m/M (m = numbers of BBs with Ilin$_i$ > 0, M = total number of BBs in a single cell; Appendix Fig. S2D). Finally, we calculated the absolute value of the mean of the

unit vector vec$_i$ ($i = 1, …, n$; $n$ = the total number of BBs) and defined Iga using the following equation:

$$\text{Iga} = \left| \frac{1}{n} \sum_{i=1}^{n} vec_i \right| * m/M * 2.5 \qquad (6)$$

If all the neighboring BBs in a single cell demonstrated V-shape alignment (Ilin$_i$ = 0), we did not find any BB-global alignment in the cell (Appendix Fig. S2D).

## Inhibitor experiments

For the depolymerization of MTs in MCCs of MTECs, 6.6 μM nocodazole/DMSO (Sigma-Aldrich) or 6.6 μM DMSO was added to the culture medium for 2 h. For the depolymerization of MTs in MCCs of the tracheal tissues, 9.9 μM nocodazole/DMSO (Sigma-Aldrich) or 9.9 μM DMSO was added to the culture medium for 2 h. The medium was changed to fresh medium to stop the reaction.

## Quantification of the angular deviation of BBs-BFs

To quantify fluctuations in the BB-BF angles within a single cell, the angular deviation of BB-BF was calculated as follows. As performed for the calculation of Io, we found the XY coordinates of BBs and BFs in a single cell from our live-cell imaging data, and we set BB/BF vectors, each of which originated from the center of a BB and ended at its accompanying BF. Then, we calculated the $\sin\theta_i$ and $\cos\theta_i$ ($i = 1, …, n$, $n$ = the total number of BBs; $0 \leq \theta_i < 360$) between each vector and the horizontal axis. Finally, we calculated the standard deviation (SD) of θ in a single cell as follows. We repeated this process for all TPs and finally calculated the mean value of the SD of $\theta_i$ in a single cell.

$$S = \sqrt{-2 ln\left( \left| \frac{1}{n} \sum_{i=1}^{n} P_i \right| \right)} \qquad (7)$$

## Quantitative analysis of the density of the apical cytoskeleton network

Calculation of the covering rate of α-tubulin and keratin8 was performed as follows using ImageJ2 Fiji.

I. Measure the area of the cell of interest

We manually cropped the cell of interest from our microscopy data reconstituted according to the maximum intensity projection at the apical region and then measured the area of the whole cell.

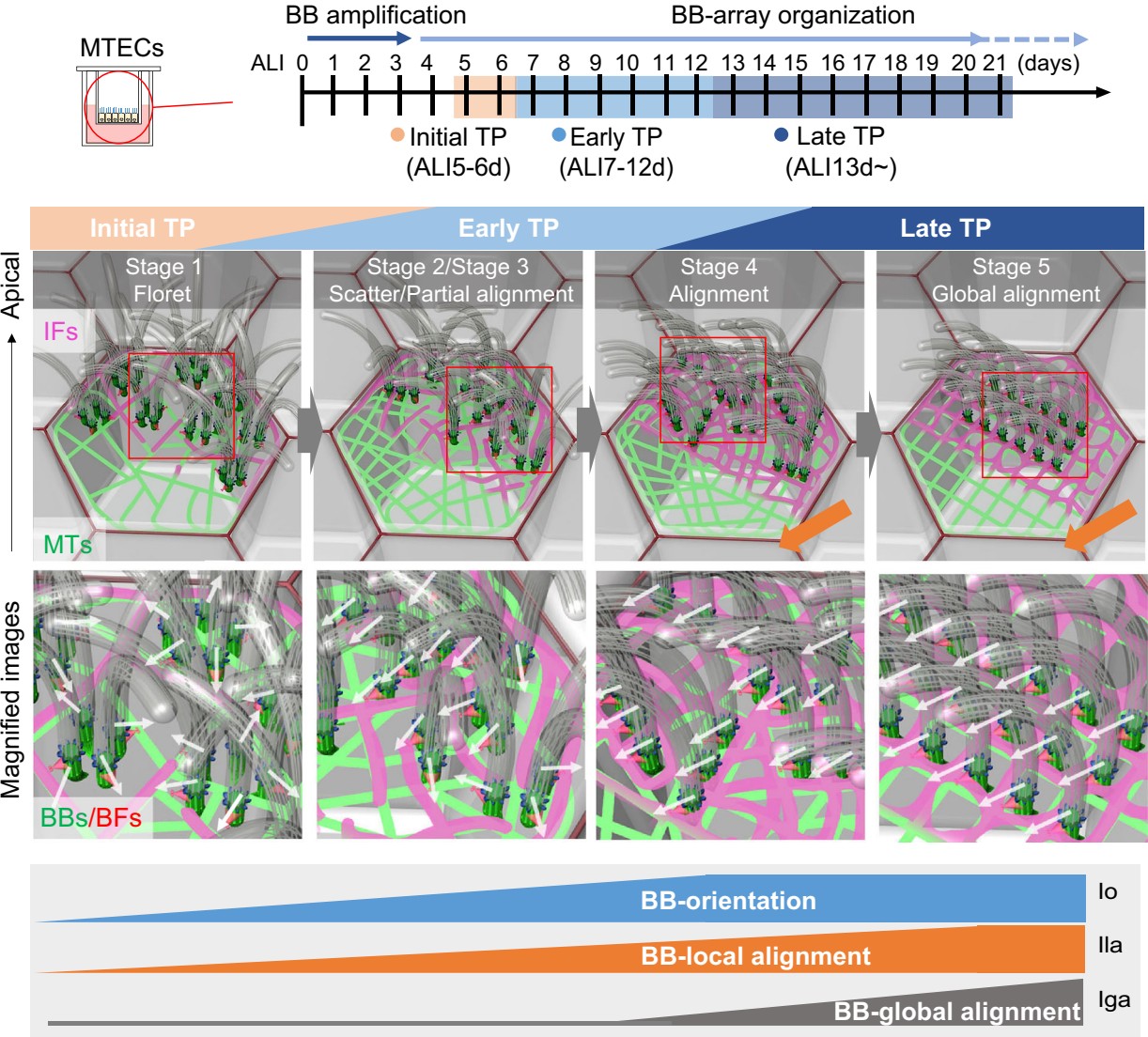

**Figure 9. Our model of the stepwise organization of the ciliary BB-array, regulated by apical cytoskeleton networks, during tracheal MCC differentiation.**

Schematic diagrams of our experimental procedures (upper diagrams) and our model of the developmental process of the BB-array, depending on the spatiotemporal relay of the apical cytoskeleton networks of MTs (green) and IFs (magenta) (lower diagrams). Schematic diagrams of cilia (gray), BBs (green), and BFs (red) are also shown.

II. Measure the binarized area of α-tubulin/keratin8

Next, we applied the tubeness filter (sigma = 0.65) in the ImageJ-plugin to the images of α-tubulin/keratin8, then binarized the images using Otsu's thresholding method. Subsequently, we measured the binarized area of α-tubulin/keratin8.

III. Calculation of the covering ratio of the apical cytoskeleton network

Finally, we calculated the covering ratio of the apical cytoskeleton from the values of the binarized area of α-tubulin/keratin8 and the area of the whole cell.

## Quantitative analysis of the lattice-like structures of apical cytoskeletons

Calculation of the SD of the area of lattice-like structures was performed as follows using ImageJ2 Fiji.

I. Generation of the image of cell shape for masking

We manually cropped the cell of interest from our microscopy data reconstituted according to the maximum intensity projection in the apical region, and we duplicated the image (Images 1 and 2). To generate an image for masking, we set the threshold (1–65535) and binarized Image 1. We filled the holes and eroded the image twice to create cell mask, after which we divided the image by 255.

II. Generation of a binarized image of the α-tubulin/keratin8 network

To generate an image for quantification, we subtracted the background (5 pix) from Image 2 and binarized the image using Li's thresholding method. Then, the image was skeletonized and inverted.

III. Calculation of the SD of the area of lattice-like structures

Finally, we multiplied Images 1 and 2 using the image calculator and acquired the binarized image of the apical cytoskeleton

network in the cell of interest. We filled the holes, eroded the image, and analyzed the particles (size: 3–400). Then, we calculated the SD of the area of binarized particles, which represented the lattice-like structures, and defined the index of α-tubulin lattice structures (Itl; 150 − SD) and the index of keratin8 lattice structures (Ikl; 120 − SD).

### Histological staining

Hematoxylin and eosin–staining and immunofluorescence staining of the mouse tissue sections were performed as described previously (Suzuki et al, 2019).

### Visualization of mucociliary clearance in the mouse trachea

To analyze mucociliary clearance, adult mouse tracheal samples were isolated from wild-type and keratin8-KO mice and observed using a fluorescence microscope and a DSU microscope (BX53-DSU; Olympus). The flow of fluorescent beads (a 500-fold dilution of Fluoresbrite, 0.5 μm; Polysciences, Inc., Warrington, PA, USA) in the mouse trachea was recorded at 25 ms/frame using a water immersion objective lens (LUMPlan FLN 60×; NA 1.00; WD, 2.0 mm; Olympus), an ORCA-Flash 4.0v2 sCMOS camera (Hamamatsu Photonics), and a ThermoPlate (37 °C; Tokai Hit). Hardware was controlled using MetaMorph software (Molecular Devices). After subtracting images processed by Gaussian Blur from all acquired images in ImageJ, each bead was tracked and analyzed using the TrackMate plugin for Imagej2 Fiji.

### Experimental design of the quantification and statistical analyses

For the quantification process, we equally observed and acquired data as much as possible, excluding the unhealthy or dying cells due to laser damage or other reasons. Moreover, to minimize subjective bias, the calculation for the Io/Ila/Iga from the XY coordinates of the BB/BFs was performed by a different person from the one who acquired the data from MCCs.

The ideal sample size for each experiment was estimated by the expected effect size. For the experiments in which significant difference was predicted from previous works, we estimated sample size using an effect size of 0.8. By contrast, for the novel experiments, we estimated sample size using an effect size of 0.5. We tried to acquire the ideal sample size of data as much as possible and confirmed the effect size of data after the experiment.

Statistical analysis was performed using Excel (Microsoft, Redmond, WA, USA), R version 4.2.1 (the R Foundation, RRID:SCR_001905), and RStudio (RStudio, PBC, RRID:SCR_000432). The normality of the data distribution was tested using the Shapiro-Wilk test, and either a parametric or nonparametric statistical method was selected. For parametric analyses, one-tailed Welch's $t$-test and one-way analysis of variance with Tukey-Kramer multiple-comparison tests were used. For nonparametric analyses, the Brunner-Munzel and Kruskal–Wallis tests with pairwise-comparison tests were used. Data are presented as means ± SDs. Results with $P$ values less than 0.05 were considered statistically significant.

## Data availability

Original images and quantification data used in this study were deposited in the BioImage Archive: accession number S-BIAD965. Any additional information about this paper is available upon request from the corresponding author.

## Peer review information

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

## Acknowledgements

We are grateful to our laboratory members for helpful discussions. We thank Fumiko Takenaga, Mariko Hata, and Miho Sawada for technical assistance. We are grateful to Dr. Toshihisa Ohtsuka (Yamanashi University, Yamanashi, Japan) for providing rabbit anti-prickle2 polyclonal antibodies. We are grateful to Drs. Tateishi and Tsutsumi for the schematic three-dimensional models of MCCs. This work was supported by the Japan Society for the Promotion of Science through a Grant-in-Aid for Specially Promoted Research (grant no. JP19H05468 to ST) and by the Core Research for Evolutionary Science and Technology program of the Japan Science and Technology Agency (grant no. JPMJCR13W4 to ST). The study was also supported by a Grant-in-Aid for Early-Career Scientists (grant no. JP18K14696 to TY), a Grant-in-Aid for Scientific Research (grant no. JP16H05121 to AT), a Grant-in-Aid for Young Scientists (B) (JP17K17853 to SK), a research grant from the Takeda Science Foundation (to ST), KOSE Cosmetology Research Foundation (to ST), a grant from the Kobayashi Foundation (to ST), and the Cooperative Research Program (Joint Usage/Research Center program) of Institute for Life and Medical Sciences, Kyoto University (to ST and GK). We would like to thank Editage (www.editage.com) for English language editing.

## Author contributions

**Gen Shiratsuchi**: Conceptualization; Data curation; Formal analysis; Validation; Investigation; Visualization; Methodology; Writing—original draft; Writing—review and editing. **Satoshi Konishi**: Conceptualization; Data curation; Formal analysis; Funding acquisition; Investigation; Visualization; Methodology; Writing—original draft. **Tomoki Yano**: Resources; Funding acquisition; Investigation. **Yuichi Yanagihashi**: Software; Methodology. **Shogo Nakayama**: Investigation; Methodology. **Tatsuya Katsuno**: Resources; Methodology. **Hiroka Kashihara**: Resources. **Hiroo Tanaka**: Resources. **Kazuto Tsukita**: Investigation. **Koya Suzuki**: Resources; Investigation. **Elisa Herawati**: Methodology. **Hitomi Watanabe**: Resources. **Toyohiro Hirai**: Supervision. **Takeshi Yagi**: Resources. **Gen Kondoh**: Resources. **Shimpei Gotoh**: Supervision. **Atsushi Tamura**: Conceptualization; Resources; Supervision; Funding acquisition; Investigation; Project administration. **Sachiko Tsukita**: Conceptualization; Supervision; Funding acquisition; Investigation; Writing—original draft; Project administration; Writing—review and editing.

## Disclosure and competing interests statement

The authors declare no competing interests.

# Expanded View Figures

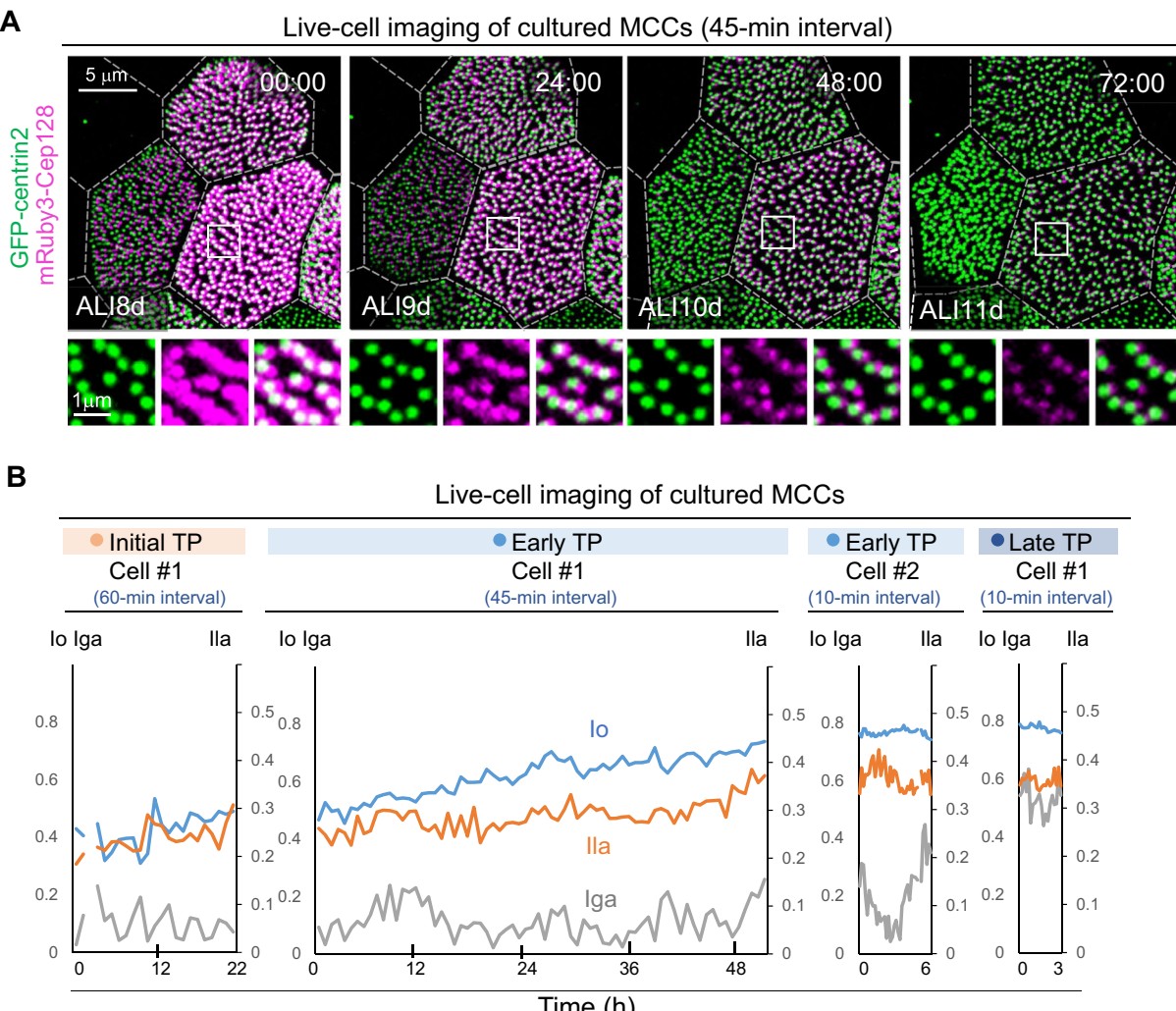

**Figure EV1.  Live-cell imaging of BB-orientation and BB-alignment in MCCs of MTECs.**

(A) Long-term dual-color live-cell imaging of MTECs prepared from transgenic mice expressing GFP-centrin2 (green) and mRuby3-Cep128 (magenta). Images were captured every 45 min. We found that mRuby3-Cep128 signals were relatively unstable compared with GFP-centrin2 signals when using long-term live-cell imaging for 3 days. Insets show 3.2-fold magnified images of fluorescent foci. Gray dotted lines represent cell shapes estimated from the background signals. Time is denoted in hours: minutes. Bars, 5 μm. See also Movie EV1. (B) Line graphs of Io (blue)/Ila (orange)/Iga (gray) calculated from the MCCs shown in Fig. 2B–E. Specifically, Initial TP Cell #1 is based on data relevant to Fig. 2B; Early TP Cell #1 on data relevant to Fig. 2C; Early TP Cell#2 on data relevant to Fig. 2D; Late TP Cell #1 on data relevant to Fig. 2E. Note that increases in Io and Ila values occurred before the increase in the Iga value. Values were plotted in the scatter plot graphs shown in Fig. 2F, G. See also Movies EV2–5. Source data are available online for this figure.

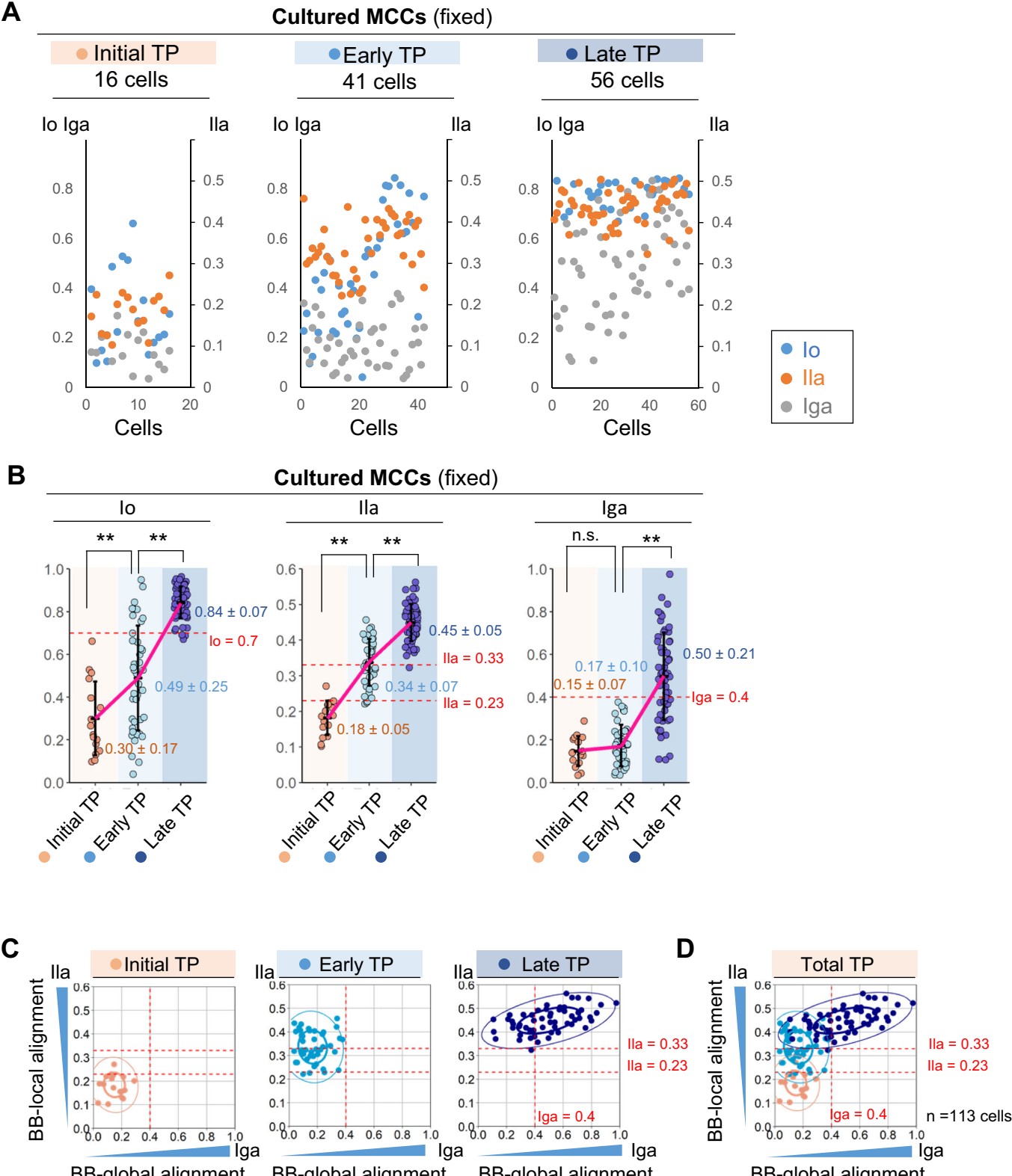

◀ **Figure EV2. Quantification of BB-orientation and BB-alignment in MCCs of fixed MTECs at different TPs.**

(A) Dot plots of Io (blue)/IIa (orange)/Iga (gray) in fixed MCCs of MTECs prepared from transgenic mice expressing GFP-centrin2 and mRuby3-Cep128 at initial ($n = 16$ cells), early ($n = 41$ cells), and late ($n = 56$ cells) timepoints (TPs). (B) Dot plots of the Io/IIa/Iga in fixed MCCs of MTECs prepared from transgenic mice expressing GFP-centrin2 and mRuby3-Cep128 at the initial (pink, $n = 16$ cells), early (sky blue, $n = 41$ cells), and late (navy blue, $n = 56$ cells) TPs. Red dotted lines represent Io = 0.7, IIa = 0.23/0.33, and Iga = 0.4. Values are means ± standard deviations of each index. \*\*$P < 0.01$, n.s., not significant (Kruskal–Wallis test with pairwise-comparison test). (C,D) Scatter plots of IIa/Iga in MCCs of MTECs prepared from transgenic mice expressing GFP-centrin2 and mRuby3-Cep128 at initial (pink, $n = 16$ cells), early (blue, $n = 41$ cells), and late (navy blue, $n = 56$ cells) TPs (C). Scatter plots from total TPs are also shown in (D). Red dotted lines represent IIa = 0.23/0.33, and Iga = 0.4. In addition, 50% (thick line) and 95% (thin line) probability ellipses are shown. Source data are available online for this figure.

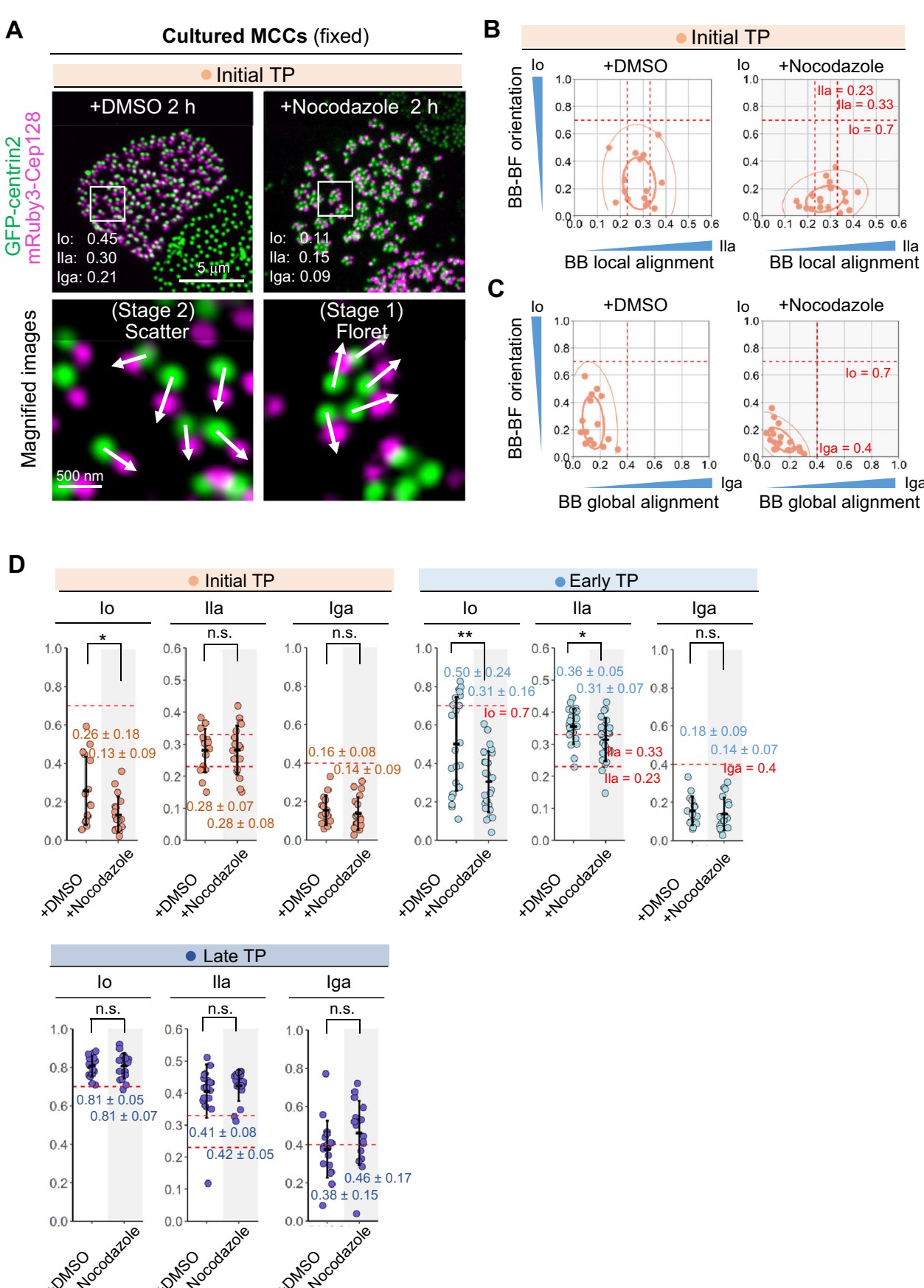

◀ **Figure EV3. Quantitative analysis of the effects of nocodazole treatment on MCCs of MTECs at different TPs.**

(A) Spinning disk confocal microscopy the BB-array in MCCs of fixed MTECs prepared from transgenic mice expressing GFP-centrin2 (green) and mRuby3-Cep128 (magenta) at the initial TP after treatment with DMSO or 6.6 μM nocodazole for 2 h. Insets show 6.7-fold magnified images of fluorescent foci. Bars, 5 μm and 500 nm. (B,C) Scatter plots of Io/IIa (B) and Io/Iga (C) in fixed MTECs prepared from transgenic mice expressing GFP-centrin2 and mRuby3-Cep128 at the initial TP after treatment with DMSO or 6.6 μM nocodazole for 2 h (DMSO, $n = 15$ cells; nocodazole, $n = 18$ cells). Red dotted lines represent Io $= 0.7$, IIa $= 0.23/0.33$, and Iga $= 0.4$. In addition, 50% (thick line) and 95% (thin line) probability ellipses are shown. (D) Dot plots of Io/IIa/Iga in MCCs of fixed MTECs prepared from transgenic mice expressing GFP-centrin2 and mRuby3-Cep128 at the at initial (pink, $n = 15$ cells and $n = 18$ cells), early (sky blue, $n = 22$ cells and 24 cells), and late (navy blue, $n = 18$ cells and $n = 17$ cells) TPs after treatment with DMSO or 6.6 μM nocodazole for 2 h. Red dotted lines represent Io $= 0.7$, IIa $= 0.23/0.33$, and Iga $= 0.4$. Values are means ± standard deviations of each index. $*P < 0.05$, $**P < 0.01$, n.s., not significant (Brunner-Munzel test). Source data are available online for this figure.

## A

### Live analysis of mucociliary transport

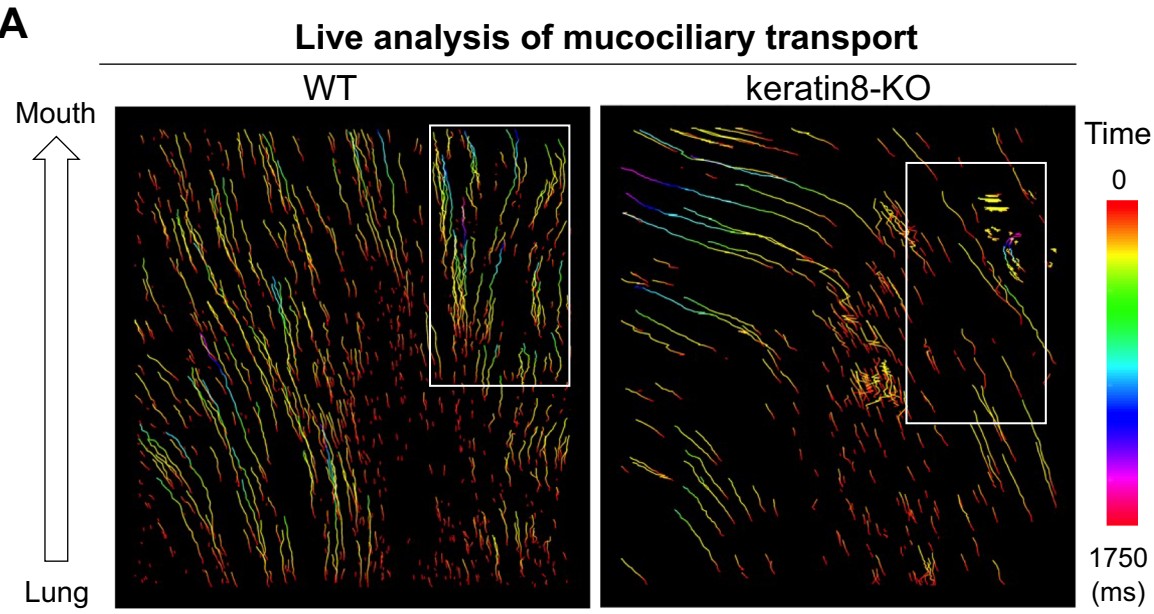

WT          keratin8-KO

Mouth

Lung

Time
0

1750
(ms)

## B

### Tracheal MCCs (fixed)

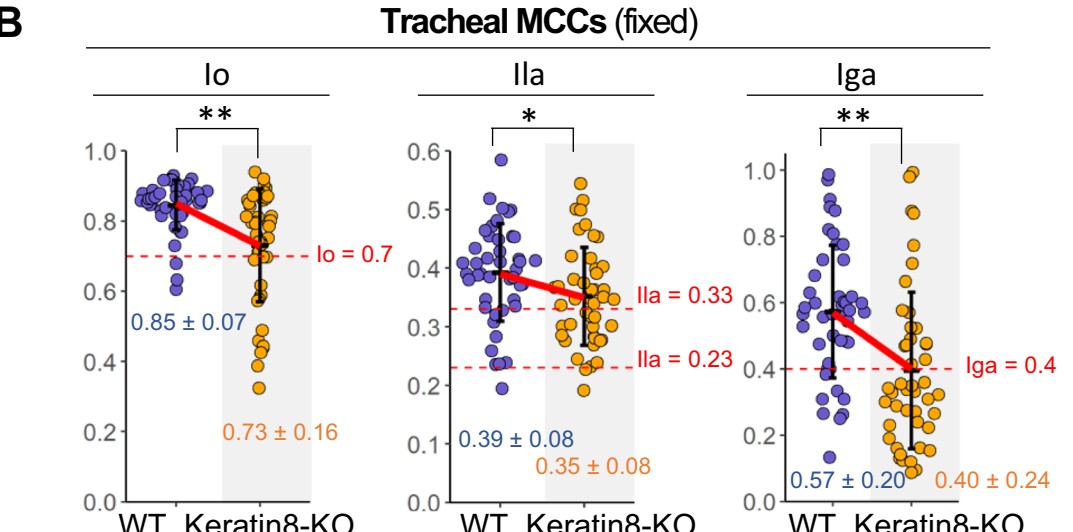

Io

**

0.85 ± 0.07

0.73 ± 0.16

Io = 0.7

WT    Keratin8-KO

IIa

*

0.39 ± 0.08
0.35 ± 0.08

IIa = 0.33
IIa = 0.23

WT    Keratin8-KO

Iga

**

0.57 ± 0.20    0.40 ± 0.24

Iga = 0.4

WT    Keratin8-KO

## C

### BB-global alignment/BB-local alignment

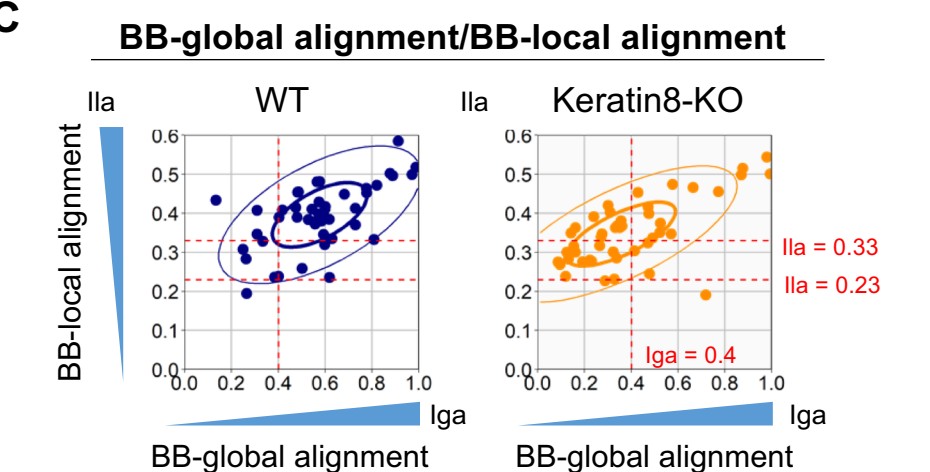

IIa          WT

BB-local alignment

IIa          Keratin8-KO

IIa = 0.33
IIa = 0.23

Iga = 0.4

Iga

BB-global alignment          BB-global alignment

    

**Figure EV4. Analysis of mucociliary clearance and the BB-array using keratin8-KO mice.**

(A) Analysis of mucociliary clearance in the tracheas of adult wild-type and keratin8-KO mice using live imaging of fluorescent beads. High-magnification images of the boxed regions are shown in Fig. 7C. (B) Dot plots of the Io/IIa/Iga in fixed MCCs of tracheal cells prepared from wild-type or keratin8-KO mice shown in Fig. 7 (wild-type, navy blue, *n* = 45 cells; keratin8-KO, orange, *n* = 45 cells). Values are means ± standard deviations of each index. \*$P < 0.05$, \*\*$P < 0.01$ (Brunner-Munzel test). Red dotted lines represent Io = 0.7, IIa = 0.23/0.33, and Iga = 0.4. (C) Scatter plots of IIa/Iga in MCCs of tracheal cells prepared from wild-type or keratin8-KO mice shown in Fig. 7 (wild-type, navy blue, *n* = 45 cells; keratin8-KO, orange, *n* = 45 cells). Red dotted lines represent IIa = 0.23/0.33, and Iga = 0.4. In addition, 50% (thick line) and 95% (thin line) probability ellipses are shown. SSource data are available online for this figure.

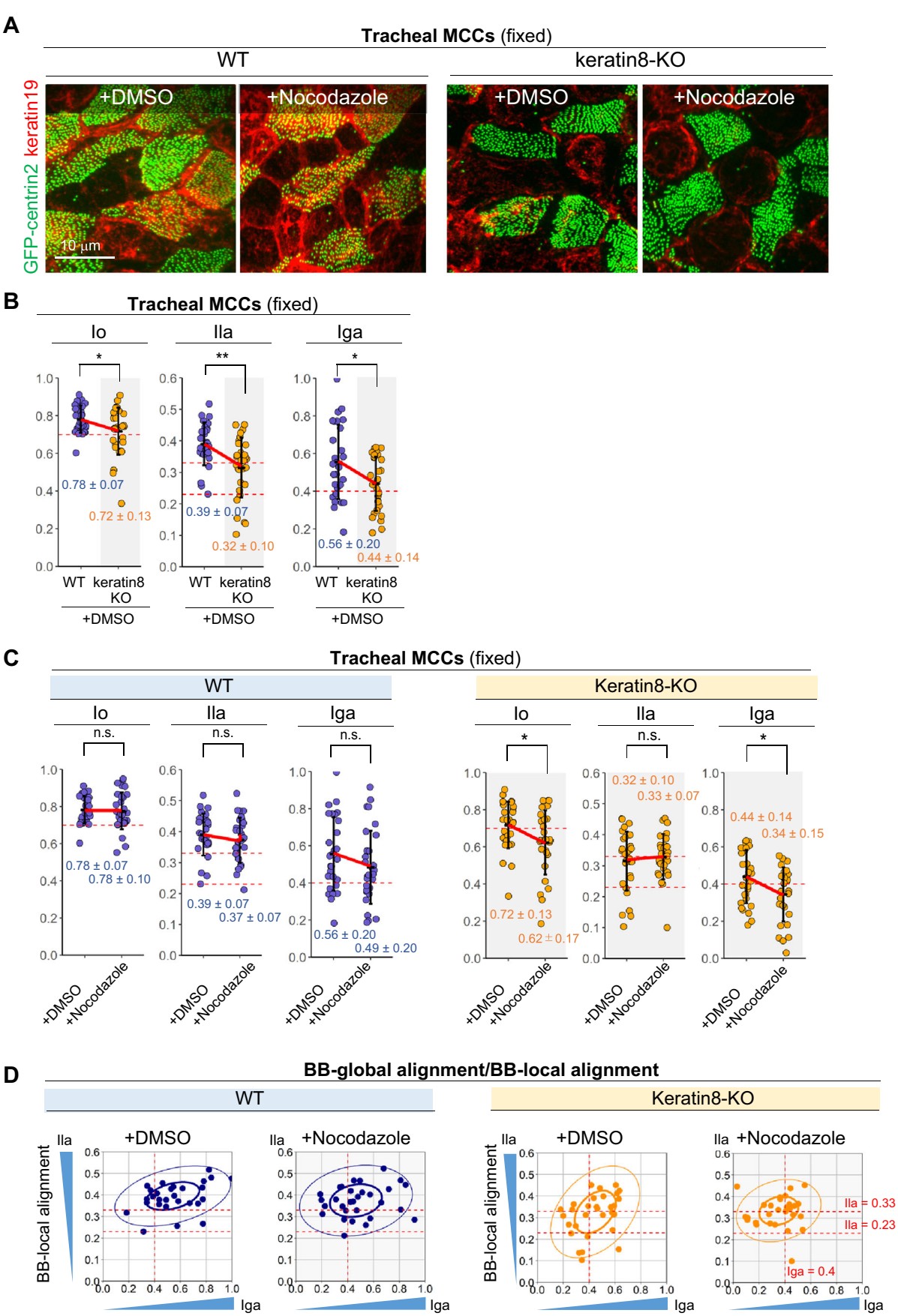

◀ **Figure EV5. Quantitative analysis of the effects of nocodazole treatment on MCCs of tracheal MTECs.**

(A) Spinning disk confocal microscopy of MCCs of fixed tracheal cells prepared from wild-type and keratin8-KO mice expressing GFP-centrin2 (green) and mRuby3-Cep128. MCCs of tracheal cells were fixed and stained using anti-α-keratin19 antibody (red) after treatment with DMSO or 9.9 μM nocodazole for 2 h. Bar, 10 μm. (B) Dot plots of the Io/IIa/Iga in fixed MCCs of tracheal cells prepared from wild-type or keratin8-KO mice after treatment with DMSO for 2 h as shown in Fig. 8 (wild-type, navy blue, $n = 28$ cells; keratin8-KO, orange, $n = 30$ cells). Values are means ± standard deviations of each index. *$P < 0.05$, **$P < 0.01$ (Brunner-Munzel test). Red dotted lines represent Io = 0.7, IIa = 0.23/0.33, and Iga = 0.4. (C) Dot plots of the Io/IIa/Iga in fixed MCCs of tracheal cells prepared from wild-type or keratin8-KO mice after treatment with DMSO or 9.9 μM nocodazole for 2 h as shown in Fig. 8 (wild-type, navy blue, $n = 28$ cells and $n = 30$ cells; keratin8-KO, orange, $n = 30$ cells and $n = 28$ cells). Values are means ± standard deviations of each index. *$P < 0.05$, **$P < 0.01$, n.s., not significant (Brunner-Munzel test). Red dotted lines represent Io = 0.7, IIa = 0.23/0.33, and Iga = 0.4. (D) Scatter plots of IIa/Iga in MCCs of tracheal cells prepared from wild-type (navy blue, $n = 28$ cells and $n = 30$ cells) or keratin8-KO (orange, $n = 30$ cells and $n = 28$ cells) mice after treatment with DMSO or 9.9 μM nocodazole for 2 h. Red dotted lines represent IIa = 0.23/0.33 and Iga = 0.4. In addition, 50% (thick line) and 95% (thin line) probability ellipses are shown. Source data are available online for this figure.

