## [Peer Review File · EMBO Reports]

Dual-color live imaging unveils stepwise organization of multiple basal body arrays by cytoskeletons

Gen Shiratsuchi, Satoshi Konishi, Tomoki Yano, Yuichi Yanagihashi, Shogo Nakayama, Tatsuya Katsuno, Hiroka Kashihara, Hiroo Tanaka, Kazuto Tsukita, Koya Suzuki, Elisa Herawati, Hitomi Watanabe, Toyohiro Hirai, Takeshi Yagi, Gen Kondoh, Shimpei Gotoh, Atsushi Tamura, and Sachiko Tsukita

DOI: [10.15252/embr.202357430](https://doi.org/10.15252/embr.202357430)

Corresponding author(s): Sachiko Tsukita (atsukita@med.teikyo-u.ac.jp), Atsushi Tamura (atamura@med.teikyo-u.ac.jp)

Review Timeline:

Transfer Date:	3rd May 23
Editorial Decision:	9th May 23
Revision Received:	29th Sep 23
Editorial Decision:	9th Nov 23
Revision Received:	13th Dec 23
Accepted:	15th Dec 23

Editor: Deniz Senyilmaz Tiebe

Transaction Report: This manuscript was transferred to EMBO reports following peer review at The EMBO Journal.

Dear Sachiko,

Thank you for transferring your manuscript to EMBO Reports, which was previously reviewed at The EMBO Journal.

Having read the manuscript and the referee reports, I would like to invite you to submit a revised manuscript to EMBO Reports as my colleague Ieva mentioned in her previous letter. In particular,

- The concerns regarding the image analysis and statistical calculations need to be addressed (referee #1 comments 1-3, referee #3 major comment 5).
- The authors would need to provide stronger evidence regarding the involvement of MTs (referee #1 comment 4, referee #3 major comment 4).
- More detailed analysis on the effect of keratin 8 LOF on lung tissue is required (referee #1, comment 6).
- Usage of the Cep128 reporter need to be better justified (referee #2, point 4).

Addressing the concerns of referee #3 regarding the mechanism of how basal feet are relevant to basal body alignment is not required for publication in EMBO Reports.

Please revise your manuscript with the understanding that the referee concerns (as in their reports) must be fully addressed and their suggestions taken on board. Please address all referee concerns in a complete point-by-point response. Acceptance of the manuscript will depend on a positive outcome of a second round of review. It is EMBO reports policy to allow a single round of major experimental revision only and acceptance or rejection of the manuscript will therefore depend on the completeness of your responses included in the next, final version of the manuscript.

We realize that it is difficult to revise to a specific deadline. In the interest of protecting the conceptual advance provided by the work, we recommend a revision within 3 months. Please discuss the revision progress ahead of this time with me if you require more time to complete the revisions, or if you have questions or comments regarding the revision (also by video chat).

1. A data availability section providing access to data deposited in public databases is missing (where applicable).
2. Your manuscript contains statistics and error bars based on $n=2$. Please use scatter plots in these cases.

You can submit the revision either as a Scientific Report or as a Research Article. For Scientific Reports, the revised manuscript can contain up to 5 main figures and 5 Expanded View figures, and it should not exceed 27000 characters. If the revision leads to a manuscript with more than 5 main figures it will be published as a Research Article. In this case the Results and Discussion section should be separate. If a Scientific Report is submitted, these sections have to be combined. This will help to shorten the manuscript text by eliminating some redundancy that is inevitable when discussing the same experiments twice. In either case, all materials and methods should be included in the main manuscript file.

4) a .docx formatted letter INCLUDING the reviewers' reports and your detailed point-by-point responses to their comments. As part of the EMBO publication's Transparent Editorial Process, EMBO reports publishes online a Review Process File (RPF) to accompany accepted manuscripts. This File will be published in conjunction with your paper and will include the referee reports, your point-by-point response and all pertinent correspondence relating to the manuscript.

<https://www.embopress.org/page/journal/14693178/authorguide#transparentprocess>

5) a complete author checklist, which you can download from our author guidelines

<https://www.embopress.org/page/journal/14693178/authorguide>. Please insert information in the checklist that is also reflected in the manuscript. The completed author checklist will also be part of the RPF.

6) Please note that all corresponding authors are required to supply an ORCID ID for their name upon submission of a revised manuscript (<<https://orcid.org/>>). Please find instructions on how to link your ORCID ID to your account in our manuscript tracking system in our Author guidelines

<<https://www.embopress.org/page/journal/14693178/authorguide#authorshipguidelines>>

7) Before submitting your revision, primary datasets produced in this study need to be deposited in an appropriate public database (see <https://www.embopress.org/page/journal/14693178/authorguide#datadeposition>). Please remember to provide a reviewer password if the datasets are not yet public. The accession numbers and database should be listed in a formal "Data Availability" section placed after Materials & Method (see also

<https://www.embopress.org/page/journal/14693178/authorguide#datadeposition>). Please note that the Data Availability Section is restricted to new primary data that are part of this study. * Note - All links should resolve to a page where the data can be accessed. *

Additional information on source data and instruction on how to label the files are available:

<https://www.embopress.org/page/journal/14693178/authorguide#sourcedata>

9) Our journal encourages inclusion of *data citations in the reference list* to directly cite datasets that were re-used and obtained from public databases. Data citations in the article text are distinct from normal bibliographical citations and should directly link to the database records from which the data can be accessed. In the main text, data citations are formatted as follows: "Data ref: Smith et al, 2001" or "Data ref: NCBI Sequence Read Archive PRJNA342805, 2017". In the Reference list, data citations must be labeled with "[DATASET]". A data reference must provide the database name, accession number/identifiers and a resolvable link to the landing page from which the data can be accessed at the end of the reference. Further instructions are available at <http://www.embopress.org/page/journal/14693178/authorguide#referencesformat>

10) Regarding data quantification (see Figure Legends:

<https://www.embopress.org/page/journal/14693178/authorguide#figureformat>)

12) Please also note our reference format:

I look forward to seeing a revised version of your manuscript when it is ready. Please let me know if you have questions or comments regarding the revision.

Kind regards,

Deniz

Deniz Senyilmaz Tiebe, PhD
Editor
EMBO Reports

Point-by-point response to the reviewer's comments

We thank the reviewers for their careful reading of our manuscript and constructive comments, which we have addressed in full, as detailed below. Accordingly, we have altered the manuscript in a substantial manner.

Referee #1:

The manuscript from Shiratsuchi utilizes long term imaging of trachea and ALI cultures to address the temporal development of basal body (BB) polarity and local and global alignment. They utilize this imaging paradigm together with detailed quantitative analysis to define a range of stages that represent different steps in the overall organization of BBs. They then go on to characterize the development of the MT and IF lattice in developing MCCs and show that short term depolymerization of MTs does not alter the IF lattice. Finally, they show that the depletion of keratin 8 leads to a decrease in flow directionality, and a delay or failure of MCCs to reach full stage 5 organization. The imaging and quantitative analysis in this paper is exquisite and the authors are to be commended. The new analysis represents an incremental advance from their previous analysis (which was also beautiful) but there are some important and subtle advances to our understanding of the process of BB alignment that come from this more detailed analysis. The authors finish the paper with a characterization of the role of IF proteins in this process and while this has the potential to be the most interesting and novel, the analysis is rather cursory. The mouse is not well characterized in this context and the analysis is not quite as rigorous as the early work in the field.

Comments:

Comment 1

In the example in Figure 2C "stage 4" there are several groups of centrioles that pretty clearly don't line up and should represent a bent line of the low IIa from the cartoon in 1G, but instead the dotted lines are straight indicating a high IIa. I feel like this representation is misleading. While I appreciate that the angles are calculated off of these lines it is confusing to see a straight line that clearly does not represent the BB alignment. A better representation of the BB alignment would be generated would be helpful.

> Thank you for your suggestions. We apologize for the confusion caused by our presenting of the results. In Figure 2, we wanted to emphasize the increase

in “BB-global alignment” and drew straight lines including 3-4 BBs to make it easy for readers to grasp our message. However, values of Ila were still calculated using the neighboring three BBs in this and other figures and schematics. We were concerned that drawing all the connections between BBs that were used for the calculation of Ila would make the figure confusing for the reader. However, we do agree with your comment that these discrepancies may be confusing.

To avoid confusion in the future, we have changed the drawing of lines in Figure 2 and the description in our revised manuscript (page 41). We drew white dotted lines between neighboring BBs which were used for the calculation of Ila, except for the BBs aligned in a V-shape that showed low Ila values. Then we demonstrated the increase in BB-global alignment as the increase of aligned shorter lines of BBs, instead of using long white dotted lines. Please note that Ila shown in figures is a mean value of the Ila calculated from individual BB pairs, in a similar fashion to the ones shown in our previous study (Herawati et al., 2016, J Cell Biol.).

Comment 2

In figure 2C there is a shift from "Partial alignment" to "alignment" that as far as I can tell is driven by a small shift in the Ila from 0.33 to 0.35 and an Io change from 0.68 to 0.7. I would like to see the statistics on these changes before I feel comfortable agreeing that these are real changes worthy of distinction. Additionally the authors define an Io of 0.7 as a delineating mark, but this seems arbitrary and no justification is made.

> Thank you for your insight. In our live-imaging observation, we showed that the value of Io was increasing during the initial/early timepoints and reached a plateau at the late timepoints in the MCCs of MTECs (Fig EV1 in the revised manuscript (Fig EV3 in the old version)). Consistently, observation of fixed MCCs also confirmed that MCCs at late timepoints showed a high degree of Io (Fig 3), and the 95% confidence interval for the Io in fixed MCCs at late timepoints was 0.7 and 1.0 (n= 56 cells, Fig 3 and Fig EV2 in the revised manuscript (Fig EV4 in the old version)). Moreover, in our previous study we used fixed MTECs and also reported a rapid increase in the Io around ALI6-7d MCCs (from Io = 0.3 to Io = 0.7) (Herawati et al., 2016, J Cell Biol.). Then we thought it would be reasonable to separate the groups of MCCs based on the

degrees of Io , and defined $Io = 0.7$ as a delineating mark to select the cells of relatively mature MCCs, including the “alignment” and “global alignment” states of BBs which we defined in this study. Otherwise, we mainly followed the definition of BB alignment as stated in Herawati et al., 2016 ($Ila < 0.23$: floret, $0.23 \leq Ila < 0.33$: scatter, $0.33 \leq Ila$: partial alignment). We have add these descriptions regarding the classification based on statistics in the revised manuscript (page 9).

Comment 3

I think a better description of the index of tubulin and keratin lattice would be helpful. It was not clear to me from the description exactly how much image manipulation was affecting this calculation. It would be nice to see a stepwise progression in supplemental as the description sounds a bit questionable. I find it a little hard to believe that there is not a significant change between initial and early for the tubulin lattice so I would like to see the process more clearly.

> Thank you for your suggestion. We have added the schematic diagrams of the step-by-step quantification methods for the density of the apical cytoskeleton network and lattice-like structures of the apical cytoskeletons in the revised Appendix Fig S3.

As you correctly pointed out, the difference between the regularity of the tubulin lattice pattern in the initial and early timepoints seemed to be large (Cohen's $d = 0.74$: medium effect size), and indeed, we also found a significant difference when we compared the values of the initial and early timepoints using the two-sample t-test. However, we acquired the value of $p = 0.06$ using ANOVA test among three timepoints, and finally we concluded that there is a significant difference between these samples inferred from our data.

Comment 4

"In addition, these data also suggested that lattice-like networks of apical IFs could support the BB-array in mature MCCs without apical MTs. ".I think a 2 HR treatment with Noco is not sufficient to make this claim. MT are more dynamic than IFs so it would make sense that the IFs would be more resistant, but could they support the BB array long term...I doubt it. I don't think these results justify this strong of a claim.

> Thank you for your comments and insight. We have confirmed the drastic decrease of tubulin signals in nocodazole-treated MCCs (6.6uM for 2 hours) (revised Appendix Fig S4 (Fig EV5 in the old version) and in our previous work (Herawati et al., 2016, J Cell Biol.)) and we think that this experiment suggests the difference in the behaviors between apical cytoskeletal filaments under the experimental conditions imposed here. But indeed, as you pointed out, we cannot say that IFs support BB alignment without apical MTs throughout MCC development in the long term, based on these data alone. Therefore, we changed the description regarding this experiment (Page 13-14).

In addition, to confirm our message, we also performed a new additional experiment using keratin8-KO mice and nocodazole-treatment (Fig 8 and EV5 in the revised manuscript). We stained tracheal MCCs of wild-type and keratin8-KO mice with DMSO or nocodazole (9.9uM for 2 hours). We found that the loss of dynamic MTs did not affect the BB array in the wildtype tracheal MCCs as observed in MTECs, but decreased BB-orientation and BB-global alignment in keratin8-KO MCCs (Fig 8 and EV5 in the revised manuscript). These results support our idea that IF networks could retain the BB array without the presence of dynamic MTs in the short-term, and also suggest that MT networks contribute to BB array formation to some extent even in mature tracheal MCCs. We have added the description about this contribution of MT networks in the revised manuscript (Page 15-16).

Comment 5

The keratin 8 KO is not well characterized in this context. Perhaps it is elsewhere but proper citations are not provided. A cursory search revealed that depending on background it can be embryonic lethal. Better references and more detail on this would be helpful. It should be explained why keratin 19 is used to show the defect in keratin 8.

> Thank you for your comments. The null mutation of the keratin8 gene has been reported to cause mid-gestational lethality in knockout mice (Baribault et al., 1993, Genes Dev.). Note that the population of mice that escape the embryonic defect and develop to the adult stage was reported to depend on the background strains of mice, with 1.6% in the C57BL/6 strain and 9.1% in the FVB/N strain being reported as escapers (Baribault et al., 1993, Genes Dev.). We used C57BL/6JJc mice in this KO experiment and found that homozygous

animals had a low fertility, but we were still able to obtain escapers that developed to the adult stage.

Previous studies have also revealed that organ-specific IF filament formation by other keratin subtypes including keratin7/19 can compensate for the loss-of-function mutation in keratin8 knockout mice and those mice can develop to the adult stage (Ameen et al., 2001, J Cell Sci., Oriolo et al., 2007, Mol Biol Cell). Therefore, in Oriolo et al., 2007, to find the intestinal epithelial cells that lacked IF filaments, the authors stained the epithelial cells in keratin8 knockout mice using pan-keratin and keratin19 antibodies. IF network formation in tracheal epithelial cells in keratin8 knockout mice has not been examined yet. Then in this work, we stained those cells using keratin19 antibody (TROMA-III; DSHB), which showed a strong signal intensity in normal tracheal tissues and examined the existence of IF filaments. We added this information in the revised manuscript (page 13) alongside the relevant references.

Comment 6

Keratin 8 has reported roles in lung function and it is not clear if the loss of flow is due to the subtle loss of polarity or some broader issue. It would be nice to see staining of the tissue to see it appears normal and that there is not a differential loss in MCCs.

>Thank you for your insight. As you pointed out, Kim et al., 2021, J Cell Sci. have reported that keratin8 loss-of-function mutation that were related to the defects of liver-caused L-arginine-mediated acute lung injury and bleomycin-induced chronic lung injury in keratin8 loss-of-function mutant mice. However, we could not find the phenotype in homeostatic alveolar tissues in keratin8-KO mice. Then we observed the alveolar tissues of the wildtype and keratin8-KO mice expressing GFP-centrin2 and mRuby3-Cep128 that were used for additional nocodazole-treatment assay (Appendix Fig S5 and p.13 in the revised manuscript). In contrast to the tracheal tissues, we found no significant morphological differences between the alveolar tissues of the wildtype and keratin8-KO mice (Appendix Fig S5 in the revised manuscript). Interestingly, we found that keratin8 and keratin18 localizations were impaired but keratin19 remained localized in the alveolar tissues of keratin8 KO mice (Appendix Fig S5 in the revised manuscript). These results suggest that keratin19 might compensate for the IF networks in the alveolar tissues of

keratin8-KO mice. We have added the description of these results in the manuscript (page 13).

Comment 7

The authors spend a significant amount of time and space highlighting the importance of their dual color centrin-cep128 imaging paradigm for scoring I_o, I_{IIa}, I_{ga}. However, when it comes to the novel findings of this paper namely the function of IFs they revert to Odf2 and centriolin which is clearly not as accurate. The whole premise of their analysis seems to hinge on them being able to discern the orientation based on a lack of staining in what is presumably the center of the BB. Please acknowledge this limitation and discrepancy.

> Thank you for your comments. In this work, as well as in our previous one (Herawati et al., 2016, J Cell Biol.), we used a custom-made program on the MATLAB software that could automatically detect the center of fluorescence signals (fluorescent dots or circles) and acquire the xy-coordinates of 100~300 BB/BF in each cell. Then, we thought that the difference between the dots and circle was not critical for the calculation of indexes. However, changing the staining/detecting condition should be apparently noted, and we added the decryption in revised manuscript (page 14).

In addition, we used keratin8-KO mice expressing GFP-centin2 and mRuby3-Cep128 for additional experiments and calculated I_o/I_{IIa}/I_{ga} from centrin/Cep128 (Fig 8 and Fig EV5 in the revised manuscript). These indexes were similar to the numbers of indexes in Fig 7, and then we concluded that the quantification using Odf2/centriolin did not significantly affect the quantification process.

Minor:

Minor comment 1

Cilia beat in the direction from BBs to BFs. Thus, the BB-orientation, determined by the direction of BBs to BFs, defines the direction of ciliary beating...This is a circular argument. I know what you are trying to say but the wording should be made more clear.

> Thank you for pointing this out. We apologize for the confusing description and have revised it accordingly (page 3).

Minor comment 2

The introduction is very well written and covers a lot of material and provides a wonderful summary of the field and the process of BB polarity. However, it seemed a bit odd to leave out one major component that is fairly well established, namely the role of fluid flow in polarization. While I appreciate it is not the focus of this paper, I think it is worthwhile to mention for the continuity of thinking about this complex process.

> Thank you for your comment. We agree with your advice and have mentioned this interesting mechanism at the epithelial and epidermal cells in the introduction section of our revised manuscript (page 4). We have also added the relevant references.

Referee #2:

The authors employed transgenic reporter lines for basal bodies (BB) and basal feet (BF) to characterize the stage-specific basal body array by calculating a mathematical index for the BB arrays based on distance, angles, and directionality inferred from BF positions. The analysis also encompasses an evaluation to determine local and global alignment derived from the intracellular orientation of BBs. The relevance of the BB array to the disorganized apical cytoskeletal network was demonstrated through the use of Krt8 knockout mice and nocodazole experiments. High-quality imaging was achieved using spinning disc super-resolution analysis, and the resulting indices provided robust and novel scientific evidence for the BB array organization. The sophisticated system and striking images captivated this reviewer, making the manuscript enjoyable to read. The mathematical analysis, in particular, can serve as a landmark for field researchers. However, several concerns were noted regarding potential improvements to the manuscript, particularly the inclusion of disease-relevant studies to enhance the broader significance of this research.

Point 1

1. This reviewer strongly suggests enhancing the clarity of the distinctions between the previously reported BB model and the two-color model described on Page 5. Emphasizing the importance of the current model using simple language will better engage readers (highlighting what is possible in this model compared to the previous one). The abstract and Page 5 contain somewhat disorganized and confusing descriptions (possibly due to excessive information).

> Thank you for your kind advice. In our previous model described in Herawati et al., 2016, J Cell Biol., we do not distinguish between BB-local and BB-global alignment. However, in this work, we have included the term BB-global alignment and demonstrated the bimodal biphasic developmental patterns of the BB-array that were masked by the previous model. Our new dual-color live-imaging system could show the deference of in fluctuation of the BB-BF during this process, which could not be observed in fixed MCCs before we added the advantage of dual-color live imaging system (page 5). We apologize

for the confusion caused by our wording in the abstract and Introduction. We have changed the description accordingly (pages 2 and 5).

Point 2

2. To elevate the importance of this study, the authors should emphasize the relevance of the Krt8 models or intermediate filaments to diseases. If no disease associations exist, incorporating an additional model may significantly underscore the significance of this research.

> Thank you for your advice. Human keratin8/18 were localized to the epithelial cells in various tissues, and mutations in keratin8 have been reported to be risk factors for liver (Strnad et al., 2008, Histochem Cell Biol.) and intestine (Polari et al., 2020, Int J Biochem Cell Biol.) diseases. Previous work has also reported L-arginine-mediated acute lung injury and bleomycin-induced chronic lung injury in keratin8 loss-of-function mutant mice (Kim et al., 2021, J Cell Sci.), but further studies are required to elucidate the malfunction of the mucociliary transport in humans.

Keratin8-knockout mice are reported to show high embryonic lethality (Baribault et al., 1993, Genes Dev.), suggesting that complete loss of epithelial IF filaments in humans would also result in severe embryonic lethality. However, if partial loss of keratin8 in tracheal MCCs occurred somehow, it might be related to the malfunction of mucociliary transport system. We have added the description in our revised manuscript (page 13).

Point 3

3. The potential binding partners of BB or BF to intermediate filaments should be identified or thoroughly discussed.

> Thank you for your advice. Previous works have shown that BF contains multiple potential binding partners of both MTs and IFs. For example, trichoplein, a keratin filament-binding protein, was shown to be localized to the centrosome/BB and required for BF-MT binding by interaction between Odf2 and ninein (Ibi et al., 2011, J Cell Sci.). The gamma-tubulin complex in BF is known to be an important factor of the BF-MT binding and mouse GCP6 is also reported to be a binding partner of epithelial keratin networks including keratin 8 *in vitro/ in vivo* (Oriolo et al., 2007, Mol Biol Cell). These results suggested that BF have the

potential to interact with MTs and keratin networks. We add the description of these potential binding partners of BF to IFs in the revised manuscript (page 18-19).

Point 4

4. Since the Cep128 reporter was used as a model system, a robust justification is necessary. Mere citations are insufficient (e.g., what is the function of this molecule? Why was it chosen for this study rather than other molecules?).

>Thank you for your comments. Cep128 has been reported to be a binding partner of Odf2 (Kashihara et al., 2019, Genes Cells) and localize to the subdistal appendages (Mazo et al., 2016, Dev Cell, Monnich et al., 2018, Cell Reports, Kashihara et al., 2019, Genes Cells) and BF of the ciliary BB (Nguyen et al., 2020, Dev Cell). Cep128 knockout caused serious loss of other BF proteins such as centriolin, Ndel1, ninein, and Cep170 (Mazo et al., 2016, Dev Cell, Kashihara et al., 2019, Genes Cells), which strongly suggests that Cep128 is essential for functional BF formation. We have added the description for Cep128 in the revised manuscript (page 6).

Point 5

5. The number of studies showed the flower-like BB structure is associated with deuterosome formation, which appears to overlap the initial stage described by the authors. They should clarify which stage correlates with these studies, including the emergence of deuterosomes.

>We apologize for any confusion. As you correctly pointed out, flower-like BB structure was reported to be associated with deuterosome formation, and this characteristic BB amplification generally occurred at ALI1-3d MTECs (Zhao et al., 2013, Nat Cell Biol.). In this study, we focused on and observed apical, membrane-attached BBs in MTECs after ALI5-6d. Previous studies have shown that these MTECs contain BBs at the last stage of amplification, and the increase in BB numbers stopped (Zhao et al., 2013, Nat Cell Biol.). Therefore, we assumed that the initial stage of BB alignment in this study is not overlapping with the BB amplification processes. To avoid confusion, we have added the explanation of deuterosome and changed the relevant description in page 8 .

Point 6

6. Related to the question above, this study lacks information on the "basolateral" centrins (centrioles) at the stage of cilia assembly initiation. Since BB assembly is the sequential event from the deuterosome-centriole (centrin) formation, this is very important. As you know, the initial stage of motile ciliogenesis includes a range of cells exhibiting distinct centriole assembly dynamics (not just flower-like structures), with centrins localized in both apical and basolateral regions of ciliating cells. The authors should describe their initial cell selection process in the materials, methods, and manuscript. Otherwise, the initial stage should be re-evaluated.

> Thank you for your advice. As you pointed out, MTECs at the initial stage of BB amplification ALI1-3d, contain many variations of BBs, and the number of apical BBs could also be highly variable. In our work, to compare the apical BB alignment in each developmental stage, we used ALI5-6d MTECs as the initial stage and focused only on the BBs at the apical membrane. We have added these descriptions in the revised manuscript (page 8).

Referee #3:

Summary:

This paper investigates the mechanism by which Basal Bodies (BB) get aligned in Multiciliated cells (MCC) of Mouse tracheal epithelia. In their previous studies, the authors had characterised the BB arrangement through a set of parameters (BB orientation index and BB alignment index) to describe the different organisational stages of BB arrangement during the initial time points of MCC differentiation. In this paper, the authors use similar parameters to understand BB arrangement, but for the entirety of the MCC differentiation process, using live imaging that spans from initial to final stages. The BB organisations were characterised using double-transgenic mice with labelling for the proteins corresponding to BB and its appendage, Basal Feet (BF). By imaging these proteins at certain crucial stages of MCC differentiation at different time intervals, the authors claim that BB orientation and BB local alignment are initially correlated which subsequently also leads to global alignment of BBs. Further, the authors claim Microtubules and Keratin IF filaments to be the regulating factors of BB alignment in the initial and final stages of MCC differentiation, respectively.

Impact:

Detailed characterisation of BB alignment process using mouse model is a valuable addition to the existing understanding of the field. The sequential illustration of BB alignment is interesting and might lead to future studies that will focus on each of these stages of BB alignment throwing light on the evolving roles of different proteins across stages. One of the major claims of the paper is the double-transgenic mice model for BB and BF protein which in future can be used for various studies by other groups. The quality of images are excellent. The figures are well illustrated and guided with schematics although the writing could be improved by avoiding long sentences. The editor may choose to publish this paper in EMBO reports provided the authors satisfactorily answer all the questions below.

Major comments:

Major comment 1

In their previous works, the authors already show that loss of BF leads to BB misalignment (Herawati et al, 2016, JCB). Having said this, one expects a well defined mechanism of how BF is relevant to BB alignment in this paper. But it is difficult to understand which experiment exactly clarifies this question.

> Thank you for your comments. Elucidation of a molecular mechanism of how BF is relevant to the BB alignment is a quite interesting issue. In this work, we shed light on the functions of apical MT and IF filaments in the early and late timepoints of BB array organization in mouse tracheal MCCs. We would appreciate your understanding that our findings would be a key to understanding these molecular mechanisms while also benefiting further studies. Following your comment, we have added the description of the predicted function of BF and potential binding partners between BFs and apical cytoskeletons in the Discussion section (page 18-19).

Major comment2

Can the BB orientation and BB alignment be sequential but independent events? Because the experimental evidence to show the correlation between the BB orientation and BB alignment is only based on the quantification of respective indices but doesn't explain the mechanism.

> Thank you for your comments. It is challenging to separately analyze BB orientation and BB-local alignment, because both events occur simultaneously in the early timepoints of MCC development and are molecularly correlated with each other through apical microtubule networks. Our previous study, where we used an Odf2 mutant that lacked BF showed a disturbed BB orientation and abnormal BB-local alignment, probably because BF has a role in BB-microtubule-binding, that is important for BB orientation, and MTOC function, that is essential for BB-local alignment, (Herawati et al., 2016, J Cell Biol.).

However, our experiments here where we used keratin8-KO mice (Fig 7) and the additional experiments including keratin8-KO mice and nocodazole treatment (Fig 8 in the revised manuscript) demonstrated that the BB-orientation and local/global BB-alignment were differently affected by the loss of MT and IF networks. Nocodazole-treated mature tracheal MCCs of keratin8-KO mice showed a decreased BB-orientation but not BB-local alignment (Fig 8 and EV5 in the revised manuscript). Therefore, we think that the BB-orientation and

BB-alignment could be independent events. We added the descriptions in the Discussion of our revised manuscript (page 18-19).

Major comment3

In the keratin8-KO mice, the MCCs show proper BB-orientation but the global alignment is disturbed. Doesn't this weaken the claim that BB-orientation and BB global alignment are correlated?

>Thank you for your comments. Referring to our response for Major comment 2, we hypothesized that the BB-orientation and BB-global alignment that occurs at late timepoints could be independent events. BB global alignment was gradually achieved after that BB orientation and BB local alignment have already been established at late timepoints. Then it would not be strange that the MCCs that could not retain BB-global alignment retrogradely moved from stage 5 to stage 4. To avoid confusion, we have changed the description in the Results section and discussed the relationships between BB-orientation and alignment in the Discussion section (page 18-19).

Major comment4

The fact that 31 % of the cells still managed to form proper BB global alignment in spite of keratin8-KO means that there could be an additional factor that might be controlling the alignment of BBs? Given the fact that Microtubules are not disrupted in this experiment, could it be microtubules? What will happen if MTs are simultaneously inhibited?

>Thank you for your comments. Following your comment, we performed additional experiments using keratin8-KO mice expressing GFP-centrin2 and mRuby3-Cep128 with nocodazole treatment (Fig 8 and EV5 in the revised manuscript). As you expected, BB-global alignment and population of stage 5 cells were significantly decreased after nocodazole treatment in keratin8-KO MCCs but not wildtype MCCs. These results suggested that MT networks also contributed to the BB-array stabilization even in mature tracheal MCCs. We have added the description in our revised manuscript (page 15-16)

Major comment5

The Early TP plot in Fig. 3D and the Total TP plot in Fig. 3E show a very different

position of the navy-blue clusters when compared to the live-cell imaging analysis in Fig. 2G. The positioning in Fig. 3D & Fig. 3E is below the Io of 0.7.

>Thank you for your comment. While the transition from stage 1 to stage 4 regularly and rapidly occurred at ALI5-7d MTECs (Herawati et al., 2016, J Cell Biol.), by contrast, the transition from stage 4 to stage 5 would occur during ALI7-14d. Because of the restriction of the recording time of live-imaging for enough fluorescence signals (~3 d), live imaging of the transition from stage 4 to stage 5 is quite hard to grasp. We acquired excellent live-cell images of ALI7 MTEC and used this cell as the early TP#2 cell in Fig 2 but this cell might not represent the major population of ALI7 cells shown in Fig 3.

Major comment6

During the evolution of keratin8 morphology, the density seems to increase from initial to early TP, however the lattice is formed only in the late TP. Why the lattice formation is not triggered while the density is increasing (Fig 5E&F) - especially given the fact that MTs are forming proper lattices?

>Thank you for your comment. We have added the schematic diagrams for the quantification in the Appendix Fig S3 in the revised manuscript. Please note that we only quantified the degrees of ordered apical lattice-like structures on the same planes of BB/BF, and uneven apical lattice structures and unstructured filaments were not counted in this method. Therefore, we think that keratin8 density might not be strictly correlated to the quantification of the lattice-like structure formation.

Major comment7

The density of prickle2 on nocodazole treatment in early TP seems reduced when compared to control in Fig EV5. But the authors claim no difference. Especially when this could also be affecting BB orientation.

>Thank you for your comment. We calculated the signal intensity of prickle2 in early/late TP samples. While the mean signal intensity of prickle2 seemed to be reduced after nocodazole treatment, there was no significant differences between the DMSO and nocodazole-treated cells. Therefore, we concluded that the reduction of prickle2 was not critical.

Minor comments:

Minor comment1

What controls the BB orientation in the context of Microtubules and Keratins? It has been shown in the Xenopus MCCs that Focal adhesion proteins (Antoniades et al., 2014, Dev. Cell) play a role in the orientation of BBs through their link to actin network. How can this be related to the current hypothesis of correlated BB orientation and BB alignment?

> Thank you for your comment. In the *Xenopus* epidermal MCCs, two distinct pools of apical actin filaments have been reported; the meshwork-like apical actin networks and subapical actin filaments that interact with the striated rootlets of BB (Werner et al., 2011, J Cell Biol.). As you pointed out, these networks were reported to be important for proper cilia spacing and global distribution (Werner et al., 2011, J Cell Biol.) and BB-localizing focal adhesion proteins, which forms the Ciliary Adhesion (CA) complex, are essential for their connections (Antoniades et al., 2014, Dev. Cell). Recent work showed an MT-dependent CA complex localization at the distal end of the BF region (Chatzifrangkeskou and Skourides 2022, Sci. Rep.), suggesting a crosstalk between apical actin and MT networks at the BF region.

By contrast, in mouse tracheal MCCs, the distribution of apical actin filaments was significantly different from that in *Xenopus* epidermal MCCs and alternatively, three distinct pools of actin filaments were demonstrated (Tateishi et al., 2017, Sci. Rep.), suggesting the functional difference of actin filaments between these two MCCs. We have already reported that cytochalasin D at a concentration that effectively disrupted the apical actin network, resulted in not only disturbed BB alignment and abnormal BB spacing, but also dissociation of BBs from the apical membrane and shrinkage of the apical surface area in tracheal MCCs (Herawati et al., 2016, J Cell Biol.). These results suggested that the actin filaments in mouse MCCs were required for MCC differentiation, but functions might be different from *Xenopus* to some extent. Instead, we added the description of regulators of apical MT and IF networks in the Discussion section (page 18-19).

Minor comment2

The BB orientation and BB alignment indices lead to stepwise classification of the process, mainly (stage 1,2,4 and 5). But it is not clear how the stage 3 (partial alignment) classification is achieved. Because this is not a specific organization as opposed to floret or scatter.

> Thank you for your comment. We apologize for the insufficient descriptions. About the classification of stages 1-3, we followed the classification method that was published in our previous work (Herawati et al., 2016, J Cell Biol.). We changed the description regarding classification (page 9) in the revised manuscript.

Minor comment3

Although the keratin8 IFs are surrounding the centrin, it looks like centrin is attached to one side of keratin lattice (Fig. 5c). What is the reason for this ?

> Thank you for your comment. Previous IF and EM observation suggested that BFs are attached to parallelly aligned apical MT filaments (Herawati et al., 2016, J Cell Biol.), and BF may move along MTs within the IF lattice-like networks and might accumulate to one side of the keratin lattice. Although we do not know the mechanism that regulates the movement of BBs towards the one side of keratin lattice-like structures, this phenomenon is plausible because of the tilted BB-orientations of MCCs being a possible reason for the mucociliary flow in a spiral pattern as observed in the human trachea (Nakamura et al., 2004, J Bronchology & Interventional Pulmonology).

Minor comment4

*There are enormous shape changes in the apical domain of the cells and it has been shown that apical domain size decides the number of BBs (in *Xenopus mucociliary epithelium*). This is apparent from Movie EV3. Shouldn't the change in apical domain taken into account for the BB global alignment?*

> Thank you for your comment. As you pointed out, our previous work showed that apical surface size was highly correlated with the number of BBs. Then we calculated the apical surface size of fixed MCCs used in Fig. 3 and examined the correlations between Iga and apical cell size. While MCCs at both early and late TPs contains cells with various apical surface size, the Iga values at the early TP were lower than those at late TP, suggesting that apical domain size were not correlated to BB-global alignment in the tracheal MCCs of mice.

Minor comment5

Are the authors using specific image processing to obtain images with almost zero background? If yes, then it will be helpful to mention all the image processing steps carried out on the raw image.

>Thank you for your comment. To avoid irradiation damage, we had to change the strength of the laser power and exposure times manually, based on the condition of MCCs. We fixed the observation conditions for each figure as far as possible, but we did not use common processing steps.

Minor comment6

In the section "BB-orientation and BB-local alignment antecedently coordinated at early timepoints, and BB-global alignment subsequently coordinated during MCC maturation", in the 3rd paragraph, why are the orientation indices are not Mentioned accurately but mentioned as a range? Proper mentioning of values will help the reader to understand easier.

>We apologize for the confusion created. We calculated the indexes from all timepoints of the live-imaging data shown in Fig. EV1 in the revised manuscript (Fig. EV3 in the old version) and only showed the cells and indexes in representative timepoints in Figure 2. To determine the stages, we considered all the timepoints and mentioned them as a range in the paragraph instead of the particular indexes shown in Figure 2.

Minor comment7

Why is 600 nm used as a threshold distance for the index of linearity of BB?

>We apologize for the lacking information. In our previous work, we mainly used 1.3 x 4 pixel (= diameter of GFP-centrin) as a threshold distance of I_{la} to exclude the second-nearest BBs from the neighboring BBs of a given BB (Herawati et al., 2016, J Cell Biol.). In our standard imaging conditions with a novel dual-color fluorescence system, 1 pixel equaled to 48.8 nm, and the diameter of GFP-centrin ranged from 7 to 9 pixel. Then we optimized the definition of a new index of liminality (I_{lin}) with a threshold distance ranging from 450 nm to 600 nm ($\approx 48.8 \times 1.3 \times 7-9$) and we selected 600 nm as a threshold distance of I_{lin} because the values of I_{lin} were highly correlated to the values of I_{la} in this condition.

Minor comment8

The writing of abstract can be improved.

> Thank you for your kind advice. To make it easy to understand our message, we revised the Abstract section (page 2).

Minor comment9

Can the authors show some experimental evidence of what happens to the BB alignment in each of these stages, especially the last stage, if and when the BB orientation is disrupted? For example, along the lines of authors previous works? (Kunimoto et al. 2012, Cell)

> Thank you for your comment. In our previous study, as you pointed out, we used the Odf2 mutant mice in which BF formation was completely disturbed in all developmental stages of the BB-array organization (Kunimoto et al., 2012, Cell). Unfortunately, we could not remove BF structures only from mature MCCs at the late TP using such kinds of BF mutants. Please note that MCCs of Odf2 mutant showed severely disturbed MT networks probably because of impaired MTOC function of the BF (Kunimoto et al., 2012, Cell, Tateishi et al., 2017, Sci Rep). Therefore, our observation of the BB alignment under inhibition of MT dynamics using nocodazole might show indirect cues for the phenotype of MCCs in which BF structures and BB orientations are disrupted. In the revised manuscript, we added the nocodazole treatment experiments using mature tracheal MCCs of the wild-type/keratin8-KO mice (page 15-16, Fig 8). In wild-type tracheal MCCs, nocodazole treatment did not affect both BB-orientation and alignment. In contrast, in keratin8-KO tracheal MCCs, nocodazole treatment decrease BB-orientation and BB-global alignment. These results suggested the possibility that disturbed BB orientation was related to a decrease in BB-global alignment when BB-surrounding IF filaments were removed.

Sincerely,

Atsushi Tamura^{1,2,3} and Sachiko Tsukita^{1,2}

¹Advanced Comprehensive Research Organization, Teikyo University, Tokyo, Japan

²Graduate School of Frontier Biosciences, Osaka University, Osaka, Japan

³School of Medicine, Teikyo University, Tokyo, Japan

<atsukita@med.teikyo-u.ac.jp> <stsukitatjcl@gmail.com>

<atamura@med.teikyo-u.ac.jp>

Dear Sachiko,

Thank you for submitting your revised manuscript. It has now been seen by all of the original referees.

As you can see, the referees find that the study is significantly improved during revision and recommend publication. However, I need you to address the points below before I can accept the manuscript.

- Please address the remaining minor concerns of referees #2 and #3.
- Please remove the Author Contribution section from the manuscript text.
- We note that the funding information has not been entered to the manuscript submission system.
- We note that Figure 8A is currently not called out in the text.
- We note that there are currently 5 Appendix figures. These Appendix figures should be placed into one pdf file with their legends placed below each figure. The Appendix legends need to be removed from the manuscript file; the Appendix figures and need to be renamed as Appendix Figure S1, etc. Please add a Table of Contents and page numbers to the Appendix files.
- The legends of the Videos need to be removed from the manuscript and should be each zipped up with their corresponding video and uploaded as one zip folder per movie. Their legends can be provided in a readme.txt file.
- 'Additional Information' paragraph should be removed from the manuscript file.
- We note a possible reuse of the same cell in Figure 8D WT+Nocodazole, which is allowed if all images are derived from the same experiment. In which case, please make a note in the figure legend.
- Please provide the BiImage Archive accession number of the mentioned source data in Data Availability section.
- As for the submitted source data, source data files need to be re-submitted as zipped folders, one .zip file for each figure. Inside each folder, the files should be organized in subfolders, one subfolder for each panel.
- Our data editors have asked you to clarify the below points in the figure legends:
 - o Please note that information related to n is missing in the legend of figure EV2b.
- Papers published in EMBO Reports include a 'synopsis' and 'bullet points' to further enhance discoverability. Both are displayed on the html version of the paper and are freely accessible to all readers. The synopsis includes a short standfirst summarizing the study in 1 or 2 sentences (max 35 words) that summarize the paper and are provided by the authors and streamlined by the handling editor. I would therefore ask you to include your synopsis blurb and 3-5 bullet points listing the key experimental findings.
- In addition, please provide an image for the synopsis. This image should provide a rapid overview of the question addressed in the study but still needs to be kept fairly modest since the image size cannot exceed 550 (width) x 300-600 (height) pixels.

Thank you again for giving us to consider your manuscript for EMBO Reports, I look forward to your minor revision.

Kind regards,

Deniz

--

Deniz Senyilmaz Tiebe, PhD
Editor
EMBO Reports

Referee #1:

The authors have adequately addressed all the comments from the initial submission.

Referee #2:

>>>Referee #2 was satisfied with the overall response from the authors.

Referee #2:

The authors employed transgenic reporter lines for basal bodies (BB) and basal feet (BF) to characterize the stage-specific basal body array by calculating a mathematical index for the BB arrays based on distance, angles, and directionality inferred from BF positions. The analysis also encompasses an evaluation to determine local and global alignment derived from the intracellular orientation of BBs. The relevance of the BB array to the disorganized apical cytoskeletal network was demonstrated through the use of Krt8 knockout mice and nocodazole experiments. High-quality imaging was achieved using spinning

disc super-resolution analysis, and the resulting indices provided robust and novel scientific evidence for the BB array organization. The sophisticated system and striking images captivated this reviewer, making the manuscript enjoyable to read. The mathematical analysis, in particular, can serve as a landmark for field researchers. However, several concerns were noted regarding potential improvements to the manuscript, particularly the inclusion of disease-relevant studies to enhance the broader significance of this research.

Point 1

1. This reviewer strongly suggests enhancing the clarity of the distinctions between the previously reported BB model and the two-color model described on Page 5. Emphasizing the importance of the current model using simple language will better engage readers (highlighting what is possible in this model compared to the previous one). The abstract and Page 5 contain somewhat disorganized and confusing descriptions (possibly due to excessive information).

> Thank you for your kind advice. In our previous model described in Herawati et al., 2016, *J Cell Biol.*, we do not distinguish between BB-local and BB-global alignment. However, in this work, we have included the term BB-global alignment and demonstrated the bimodal biphasic developmental patterns of the BB-array that were masked by the previous model. Our new dual-color live-imaging system could show the deference of in fluctuation of the BB-BF during this process, which could not be observed in fixed MCCs before we added the advantage of dual-color live imaging system (page 5). We apologize for the confusion caused by our wording in the abstract and Introduction. We have changed the description accordingly (pages 2 and 5).

>>> -Satisfied.

Point 2

2. To elevate the importance of this study, the authors should emphasize the relevance of the Krt8 models or intermediate filaments to diseases. If no disease associations exist, incorporating an additional model may significantly underscore the significance of this research.

> Thank you for your advice. Human keratin8/18 were localized to the epithelial cells in various tissues, and mutations in keratin8 have been reported to be risk factors for liver (Strnad et al., 2008, *Histochem Cell Biol.*) and intestine (Polari et al., 2020, *Int J Biochem Cell Biol.*) diseases. Previous work has also reported L-arginine-mediated acute lung injury and bleomycin-induced chronic lung injury in keratin8 loss-of-function mutant mice (Kim et al., 2021, *J Cell Sci.*), but further studies are required to elucidate the malfunction of the mucociliary transport in humans.

Keratin8-knockout mice are reported to show high embryonic lethality (Baribault et al., 1993, *Genes Dev.*), suggesting that complete loss of epithelial IF filaments in humans would also results in severe embryonic lethality. However, if partial loss of keratin8 in tracheal MCCs occurred somehow, it might be related to the malfunction of mucociliary transport system. We have added the description in our revised manuscript (page 13).

>>>For more explicit justification and improved flow, the author should initially mention their prior findings related to Krt8 instead of delving directly into injury repair. Additionally, it's pertinent to note that Krt8 is among the most prevalently expressed intermediate filaments (IF) in differentiating luminal cells (Rock et al. (2009)). Moreover, Krt8 expression is observed in a subset of basal cells during airway differentiation, as evidenced by Watson et al. (2015) and Mori et al. (2015).

Point 3

3. The potential binding partners of BB or BF to intermediate filaments should be identified or thoroughly discussed.

> Thank you for your advice. Previous works have shown that BF contains multiple potential binding partners of both MTs and IFs. For example, trichoplein, a keratin filament-binding protein, was shown to be localized to the centrosome/BB and required for BF-MT binding by interaction between Odf2 and ninein (Ibi et al., 2011, *J Cell Sci.*). The gamma-tubulin complex in BF is known to be an important factor of the BF-MT binding and mouse GCP6 is also reported to be a binding partner of epithelial keratin networks including keratin 8 in vitro/ in vivo (Oriolo et al., 2007, *Mol Biol Cell*). These results suggested that BF have the potential to interact with MTs and keratin networks. We add the description of

these potential binding partners of BF to IFs in the revised manuscript (page 18-19).

>>> -Satisfied with the description on page 19-20.

Point 4

4. Since the Cep128 reporter was used as a model system, a robust justification is necessary. Mere citations are insufficient (e.g., what is the function of this molecule? Why was it chosen for this study rather than other molecules?).

> Thank you for your comments. Cep128 has been reported to be a binding partner of Odf2 (Kashihara et al., 2019, Genes Cells) and localize to the subdistal appendages (Mazo et al., 2016, Dev Cell, Monnich et al., 2018, Cell Reports, Kashihara et al., 2019, Genes Cells) and BF of the ciliary BB (Nguyen et al., 2020, Dev Cell). Cep128 knockout caused serious loss of other BF proteins such as centriolin, Ndel1, ninein, and Cep170 (Mazo et al., 2016, Dev Cell, Kashihara et al., 2019, Genes Cells), which strongly suggests that Cep128 is essential for functional BF formation. We have added the description for Cep128 in the revised manuscript (page 6).

>>> -Satisfied with the description on page 7

Point 5

5. The number of studies showed the flower-like BB structure is associated with deuterosome formation, which appears to overlap the initial stage described by the authors. They should clarify which stage correlates with these studies, including the emergence of deuterosomes.

> We apologize for any confusion. As you correctly pointed out, flower-like BB structure was reported to be associated with deuterosome formation, and this characteristic BB amplification generally occurred at ALI1-3d MTECs (Zhao et al., 2013, Nat Cell Biol.). In this study, we focused on and observed apical, membrane-attached BBs in MTECs after ALI5-6d. Previous studies have shown that these MTECs contain BBs at the last stage of amplification, and the increase in BB numbers stopped (Zhao et al., 2013, Nat Cell Biol.). Therefore, we assumed that the initial stage of BB alignment in this study is not overlapping with the BB amplification processes. To avoid confusion, we have added the explanation of deuterosome and changed the relevant description in page 8 .

>>>-Satisfied with the description on page 9.

Point 6

6. Related to the question above, this study lacks information on the "basolateral" centrins (centrioles) at the stage of cilia assembly initiation. Since BB assembly is the sequential event from the deuterosome-centriole (centrin) formation, this is very important. As you know, the initial stage of motile ciliogenesis includes a range of cells exhibiting distinct centriole assembly dynamics (not just flower-like structures), with centrins localized in both apical and basolateral regions of ciliating cells. The authors should describe their initial cell selection process in the materials, methods, and manuscript. Otherwise, the initial stage should be re-evaluated.

> Thank you for your advice. As you pointed out, MTECs at the initial stage of BB amplification ALI1-3d, contain many variations of BBs, and the number of apical BBs could also be highly variable. In our work, to compare the apical BB alignment in each developmental stage, we used ALI5-6d MTECs as the initial stage and focused only on the BBs at the apical membrane. We have added these descriptions in the revised manuscript (page 8).

>>> -Satisfied with the description on page 9.

Referee #3:

In this revised manuscript the authors have done a nice job of extending their analysis of Keratin 8 and have added some nice data. Furthermore they have addressed many of the concerns I had via changes to the texts and expanded explanations.

Overall I think this is a really beautiful study that should be published in EMBO reports. I do have one minor point that I think should be addressed.

I requested the addition of fluid flow and polarity to the introduction and I appreciate the authors attempt to do this. However, the authors present this as an open question in the field with the potential that it is tissue specific. Based on the clinical data from numerous PCD patients (over decades) I think it is pretty well established that cilia immobility affects polarity in the respiratory tract (for one example of many Biggart et al. 2001). To reference one paper that suggests otherwise when that paper did not rigorously quantify motility or flow seems misleading. Obviously, a lot of important factors go into orienting basal bodies, and it was not my intention to take away from the current findings, but only to be inclusive. However, the way it has been presented seems inaccurate based on the literature as a whole.

Point-by-point response to the comments of editor and reviewers

We thank you and the reviewers for their careful reading of our manuscript and for the constructive comments which have substantially improved it. We have addressed the comments of the editor (pages 1–5) and reviewers (page 7–8 and 10), and have accordingly revised our manuscript. Additionally, we have prepared the synopsis blurb and the bullet points summarizing our findings, as well as the Appendix Figure file (pdf), the zipped Video folders, and the zipped Source data folders. Please note that the changes to our manuscript file in this version have been highlighted in light blue. We have also made corrections to the colors and thickness of the lines, as well as the positioning of characters in Appendix Figures and modified the orientation of Movie EV9.

Dear Sachiko,

Thank you for submitting your revised manuscript. It has now been seen by all of the original referees. As you can see, the referees find that the study is significantly improved during revision and recommend publication. However, I need you to address the points below before I can accept the manuscript.

- Please address the remaining minor concerns of referees #2 and #3.*

>We addressed the concerns of referees #2 and #3 (perhaps #2 and #1 in the previous version) as follows: (page 7–8 and page 10 in this file, and page 13 and page 4 in the manuscript file).

- Please remove the Author Contribution section from the manuscript text.*

>We removed the Author Contribution section from our manuscript text (page 30).

- We note that the funding information has not been entered to the manuscript submission system.*

>We added the funding information on the manuscript submission system.

1. A Grant-in-Aid for Specially Promoted Research from the Japan Society for the Promotion of Science (JSPS) (Grant Number JP19H05468, to S. Tsukita).
2. A Core Research for Evolutionary Science and Technology (CREST) program grant from the Japan Science and Technology Agency (JST) (Grant Number JPMJCR13W4, to S. Tsukita).
3. A Grant-in-Aid for Early-Career Scientists from the Japan Society for the Promotion of Science (JSPS) (Grant Number JP18K14696, to T. Yano).
4. A Grant-in-Aid for Scientific Research from the Japan Society for the Promotion of Science (JSPS) (Grant Number JP16H05121, to A. Tamura).
5. A Grant-in-Aid for Young Scientists (B) from the Japan Society for the Promotion of Science (JSPS) (Grant Number JP17K17853, to S. Konishi).
6. A research grant from the Takeda Science Foundation (to S. Tsukita).
7. A research grant from the KOSE Cosmetology Research Foundation (to S. Tsukita).
8. A grant from the Kobayashi Foundation (to S. Tsukita).

• *We note that Figure 8A is currently not called out in the text.*

>We have now included a reference to Figure 8A in the text of our manuscript text on page 15.

• *We note that there are currently 5 Appendix figures. These Appendix figures should be placed into one pdf file with their legends placed below each figure. The Appendix legends need to be removed from the manuscript file; the Appendix figures and need to be renamed as Appendix Figure S1, etc. Please add a Table of Contents and page numbers to the Appendix files.*

>We have removed the Appendix Figure Legends from pages 47–50 of our manuscript file. Additionally, we have compiled the Appendix Figures S1–S5 with their respective legends into a single PDF file. This file also includes a Table

of Contents and page numbers for easy navigation.

- *The legends of the Videos need to be removed from the manuscript and should be each zipped up with their corresponding video and uploaded as one zip folder per movie. Their legends can be provided in a readme.txt file.*

> We have removed the legends for the Videos from pages 47–50 of our manuscript. Furthermore, we have created 11 individual zip folders, each containing the corresponding Video file and a readme.txt file with the relevant legend.

- *'Additional Information' paragraph should be removed from the manuscript file.*

> We removed 'Additional Information' paragraph from our manuscript file (page 30).

- *We note a possible reuse of the same cell in Figure 8D WT+Nocodazole, which is allowed if all images are derived from the same experiment. In which case, please make a note in the figure legend.*

> Thank you for bringing this to our attention. In Figure 8D (WT+Nocodazole), we indeed utilized two cells from a single large image. A similar approach was employed for Figure 7, where the wild-type Stage 5 cell was extracted from the larger image presented in Figure 7E/F, and keratin8-KO Stage 4 and 5 cells were selected from the large image in Figure 7F. We assure you that all images used for quantification and as representative figures in Figures 7 and 8 were sourced from the same experiment, respectively. We have verified that this selection does not impact the results of the quantitative analysis. To clarify this, we have included an explanatory note in the legends of both figures on pages 45 and 46 of our manuscript.

- *Please provide the BioImage Archive accession number of the mentioned source data in Data Availability section.*

> In response to your request, we have included the BioImage Archive accession number S-BIAD965 in the Data Availability section of our manuscript on page 29.

• As for the submitted source data, source data files need to be re-submitted as zipped folders, one .zip file for each figure. Inside each folder, the files should be organized in subfolders, one subfolder for each panel.

> We have organized the source data into 17 zipped folders corresponding to each figure, which include subfolders for individual panels as requested. These encompass eight Figures, five Expanded View Figures, and four Appendix Figures.

• Our data editors have asked you to clarify the below points in the figure legends:

o Please note that information related to n is missing in the legend of figure EV2b.

> Per the request of the data editors, we have clarified the figure legend for Figure EV2B by adding the number of plots used in the analysis. This information can be found on page 47 of our manuscript.

• Papers published in EMBO Reports include a 'synopsis' and 'bullet points' to further enhance discoverability. Both are displayed on the html version of the paper and are freely accessible to all readers. The synopsis includes a short standfirst summarizing the study in 1 or 2 sentences (max 35 words) that summarize the paper and are provided by the authors and streamlined by the handling editor. I would therefore ask you to include your synopsis blurb and 3-5 bullet points listing the key experimental findings.

> In accordance with your instructions, we have crafted a concise synopsis and included a list of three key bullet points that encapsulate the core findings of our

study. These have been added to the specified section of our manuscript for enhanced discoverability and reader engagement.

Synopsis blurb

High-resolution live-cell imaging revealed the stepwise organization of ciliary basal body (BB) arrays in mouse tracheal multiciliated cells (MCCs), regulated by the apical cytoskeleton networks.

Bullet points of our key findings

- We developed a high-resolution, dual-color, live-cell imaging system to monitor the dynamic organization of ciliary basal bodies (BBs) and basal feet (BF) in primary cultured mouse tracheal epithelial cells (MTECs).
- We have identified a stepwise organizational process for the unidirectional orientation of BB-BF and the alignment of BBs from local to global scales during the differentiation of multiciliated cells (MCCs).
- The research strongly indicated stage-specific contributions of apical microtubules (MTs) and apical intermediate filaments (IFs) to the organization of the basal body (BB) array.

• In addition, please provide an image for the synopsis. This image should provide a rapid overview of the question addressed in the study but still needs to be kept fairly modest since the image size cannot exceed 550 (width) x 300-600 (height) pixels.

> We provided the image for the synopsis of our study including our experimental strategy and the novel findings as “Synopsis” file (eps and pdf).

Referee #1 (perhaps Referee #3 in 1st revision process):

The authors have adequately addressed all the comments from the initial submission.

> We appreciate your constructive comments and suggestions, which have significantly contributed to the improvement of our manuscript. Thank you for

your thorough review and valuable insights.

Referee #2:

>>>Referee #2 was satisfied with the overall response from the authors.

Referee #2:

The authors employed transgenic reporter lines for basal bodies (BB) and basal feet (BF) to characterize the stage-specific basal body array by calculating a mathematical index for the BB arrays based on distance, angles, and directionality inferred from BF positions. The analysis also encompasses an evaluation to determine local and global alignment derived from the intracellular orientation of BBs. The relevance of the BB array to the disorganized apical cytoskeletal network was demonstrated through the use of Krt8 knockout mice and nocodazole experiments. High-quality imaging was achieved using spinning disc super-resolution analysis, and the resulting indices provided robust and novel scientific evidence for the BB array organization. The sophisticated system and striking images captivated this reviewer, making the manuscript enjoyable to read. The mathematical analysis, in particular, can serve as a landmark for field researchers. However, several concerns were noted regarding potential improvements to the manuscript, particularly the inclusion of disease-relevant studies to enhance the broader significance of this research.

Point 1

1. This reviewer strongly suggests enhancing the clarity of the distinctions between the previously reported BB model and the two-color model described on Page 5. Emphasizing the importance of the current model using simple language will better engage readers (highlighting what is possible in this model compared to the previous one). The abstract and Page 5 contain somewhat disorganized and confusing descriptions (possibly due to excessive information).

>Thank you for your kind advice. In our previous model described in Herawati et al., 2016, J Cell Biol., we do not distinguish between BB-local and BB-global alignment. However, in this work, we have included the term BB-global alignment and demonstrated the bimodal biphasic developmental patterns of the

BB-array that were masked by the previous model. Our new dual-color live-imaging system could show the deference of in fluctuation of the BB-BF during this process, which could not be observed in fixed MCCs before we added the advantage of dual-color live imaging system (page 5). We apologize for the confusion caused by our wording in the abstract and Introduction. We have changed the description accordingly (pages 2 and 5).

>>> -Satisfied.

Point 2

2. To elevate the importance of this study, the authors should emphasize the relevance of the Krt8 models or intermediate filaments to diseases. If no disease associations exist, incorporating an additional model may significantly underscore the significance of this research.

>Thank you for your advice. Human keratin8/18 were localized to the epithelial cells in various tissues, and mutations in keratin8 have been reported to be risk factors for liver (Strnad et al., 2008, Histochem Cell Biol.) and intestine (Polari et al., 2020, Int J Biochem Cell Biol.) diseases. Previous work has also reported L-arginine-mediated acute lung injury and bleomycin-induced chronic lung injury in keratin8 loss-of-function mutant mice (Kim et al., 2021, J Cell Sci.), but further studies are required to elucidate the malfunction of the mucociliary transport in humans. Keratin8-knockout mice are reported to show high embryonic lethality (Baribault et al., 1993, Genes Dev.), suggesting that complete loss of epithelial IF filaments in humans would also results in severe embryonic lethality. However, if partial loss of keratin8 in tracheal MCCs occurred somehow, it might be related to the malfunction of mucociliary transport system. We have added the description in our revised manuscript (page 13).

>>>For more explicit justification and improved flow, the author should initially mention their prior findings related to Krt8 instead of delving directly into injury repair. Additionally, it's pertinent to note that Krt8 is among the most prevalently expressed intermediate filaments (IF) in differentiating luminal cells (Rock et al. (2009)). Moreover, Krt8 expression is observed in a subset of basal cells during airway differentiation, as evidenced by Watson et al. (2015) and Mori et al. (2015).

>Thank you for your insightful comment. We concur with your suggestion and have accordingly revised our manuscript. Specifically, on page 13, we have introduced a new section detailing the expression of keratin-8 in basal luminal progenitor cells during airway epithelial differentiation, prior to discussing the phenotype of keratin8-KO mice. Furthermore, we have incorporated the relevant references to Rock et al. (2009), Watson et al. (2015), and Mori et al. (2015) on pages 35, 36, and 38, respectively, to support these additions.

Point 3

3. The potential binding partners of BB or BF to intermediate filaments should be identified or thoroughly discussed.

>Thank you for your advice. Previous works have shown that BF contains multiple potential binding partners of both MTs and IFs. For example, trichoplein, a keratin filament-binding protein, was shown to be localized to the centrosome/BB and required for BF-MT binding by interaction between Odf2 and ninein (Ibi et al., 2011, J Cell Sci.). The gamma-tubulin complex in BF is known to be an important factor of the BF-MT binding and mouse GCP6 is also reported to be a binding partner of epithelial keratin networks including keratin 8 in vitro/ in vivo (Oriolo et al., 2007, Mol Biol Cell). These results suggested that BF have the potential to interact with MTs and keratin networks. We add the description of these potential binding partners of BF to IFs in the revised manuscript (page 18-19).

>>> -Satisfied with the description on page 19-20.

Point 4

4. Since the Cep128 reporter was used as a model system, a robust justification is necessary. Mere citations are insufficient (e.g., what is the function of this molecule? Why was it chosen for this study rather than other molecules?).

>Thank you for your comments. Cep128 has been reported to be a binding partner of Odf2 (Kashihara et al., 2019, Genes Cells) and localize to the subdistal appendages (Mazo et al., 2016, Dev Cell, Monnich et al., 2018, Cell Reports, Kashihara et al., 2019, Genes Cells) and BF of the ciliary BB (Nguyen et al., 2020, Dev Cell). Cep128 knockout caused serious loss of other BF proteins such as centriolin, Ndel1, ninein, and Cep170 (Mazo et al., 2016, Dev

Cell, Kashihara et al., 2019, Genes Cells), which strongly suggests that Cep128 is essential for functional BF formation. We have added the description for Cep128 in the revised manuscript (page 6).

>>> -Satisfied with the description on page 7

Point 5

5. The number of studies showed the flower-like BB structure is associated with deuterosome formation, which appears to overlap the initial stage described by the authors. They should clarify which stage correlates with these studies, including the emergence of deuterosomes.

>We apologize for any confusion. As you correctly pointed out, flower-like BB structure was reported to be associated with deuterosome formation, and this characteristic BB amplification generally occurred at ALI1-3d MTECs (Zhao et al., 2013, Nat Cell Biol.). In this study, we focused on and observed apical, membrane-attached BBs in MTECs after ALI5-6d. Previous studies have shown that these MTECs contain BBs at the last stage of amplification, and the increase in BB numbers stopped (Zhao et al., 2013, Nat Cell Biol.). Therefore, we assumed that the initial stage of BB alignment in this study is not overlapping with the BB amplification processes. To avoid confusion, we have added the explanation of deuterosome and changed the relevant description in page 8 .

>>>-Satisfied with the description on page 9.

Point 6

6. Related to the question above, this study lacks information on the "basolateral" centrins (centrioles) at the stage of cilia assembly initiation. Since BB assembly is the sequential event from the deuterosome-centriole (centrin) formation, this is very important. As you know, the initial stage of motile ciliogenesis includes a range of cells exhibiting distinct centriole assembly dynamics (not just flower-like structures), with centrins localized in both apical and basolateral regions of ciliating cells. The authors should describe their initial cell selection process in the materials, methods, and manuscript. Otherwise, the initial stage should be re-evaluated.

>Thank you for your advice. As you pointed out, MTECs at the initial stage of

BB amplification ALI1-3d, contain many variations of BBs, and the number of apical BBs could also be highly variable. In our work, to compare the apical BB alignment in each developmental stage, we used ALI5-6d MTECs as the initial stage and focused only on the BBs at the apical membrane. We have added these descriptions in the revised manuscript (page 8).

>>> -Satisfied with the description on page 9.

Referee #3 (perhaps Referee #1 in 1st revision process):

In this revised manuscript the authors have done a nice job of extending their analysis of Keratin 8 and have added some nice data. Furthermore they have addressed many of the concerns I had via changes to the texts and expanded explanations. Overall I think this is a really beautiful study that should be published in EMBO reports. I do have one minor point that I think should be addressed.

I requested the addition of fluid flow and polarity to the introduction and I appreciate the authors attempt to do this. However, the authors present this as an open question in the field with the potential that it is tissue specific. Based on the clinical data from numerous PCD patients (over decades) I think it is pretty well established that cilia immobility affects polarity in the respiratory tract (for one example of many Biggart et al. 2001). To reference one paper that suggests otherwise when that paper did not rigorously quantify motility or flow seems misleading. Obviously, a lot of important factors go into orienting basal bodies, and it was not my intention to take away from the current findings, but only to be inclusive. However, the way it has been presented seems inaccurate based on the literature as a whole.

>Thank you for your constructive feedback. We have taken your suggestion into account and revised the section on fluid flow and polarity in the Introduction (page 4) of our manuscript. In order to present a clearer and more accurate message to our readers, we have removed the reference to the controversial report and seamlessly integrated information about fluid flow into the preceding paragraph. Furthermore, as you recommended, we have included a reference to

a clinical report on human subjects (page 4). The relevant citation has been added to our reference list (page 31).

Dear Sachiko,

Thank you for submitting your revised manuscript. I have now looked at everything and all is fine. Therefore, I am very pleased to accept your manuscript for publication in EMBO Reports.

Congratulations on a nice work!

Before we can transfer your manuscript to our production team, I need your input on the two outstanding points below:

1) The dataset with the accession number S-BIAD965 cannot be found on BiolImage Archive.

2) I made some minor changes in the bullet points to increase clarity and accessibility. Please take a look and confirm, or feel free to propose further changes.

- A high-resolution, dual-color, live-cell imaging system is developed to monitor the dynamic organization of ciliary basal bodies (BBs) and basal feet (BF) in primary cultured mouse tracheal epithelial cells (MTECs).
- The unidirectional orientation of BB-BF and the alignment of BBs from local to global scales during the differentiation of multiciliated cells (MCCs) exhibit a stepwise organizational process.
- Apical microtubules (MTs) and apical intermediate filaments (IFs) provide stage-specific contributions to the organization of the basal body (BB) array.

Thank you.

Kind regards,

Deniz

--

Deniz Senyilmaz Tiebe, PhD

Editor

EMBO Reports
